# Conceptualising a Cloud Business Intelligence Security Evaluation Framework for Small and Medium Enterprises in Small Towns of the Limpopo Province, South Africa

**Moses Moyo** * **and Marianne Loock** *

Information Systems Department, School of Computing, College of Science, Engineering, and Technology, University of South Africa (UNISA), Florida Campus, Johannesburg 1709, South Africa
* Correspondence: mosesm50@gmail.com (M.M.); loockm@unisa.ac.za (M.L.)

**Abstract:** The purpose of this study was to investigate security evaluation practices among small and medium enterprises (SMEs) in small South African towns when adopting cloud business intelligence (Cloud BI). The study employed a quantitative design in which 57 SMEs from the Limpopo Province were surveyed using an online questionnaire. The study found that: (1) the level of cybersecurity threats awareness among decision-makers was high; (2) decision-makers preferred simple checklists and guidelines over conventional security policies, standards, and frameworks; and (3) decision-makers considered financial risks, data and application security, and cloud service provider reliability as the main aspects to consider when evaluating Cloud BI applications. The study conceptualised a five-component security framework for evaluating Cloud BI applications, integrating key aspects of conventional security frameworks and methodologies. The framework was validated for relevance by IT specialists and acceptance by SME owners. The Spearman correlational test for relevance and acceptance of the proposed framework was found to be highly significant at $p < 0.05$. The study concluded that SMEs require user-friendly frameworks for evaluating Cloud BI applications. The major contribution of this study is the security evaluation framework conceptualised from the best practices of existing security standards and frameworks for use by decision-makers from small towns in Limpopo. The study recommends that future research consider end-user needs when customising or proposing new solutions for SMEs in small towns.

**Keywords:** cloud business intelligence; small and medium enterprise; security evaluation framework; security vulnerabilities; security risk

## 1. Introduction

Cloud services play an important role in the social and economic sectors in South Africa [1,2]. In the past few years, small and medium enterprises (SMEs) have not been able to utilise on-premises business solutions such as business intelligence (BI), customer relations management, and enterprise resource planning. This is due to the limited financial resources needed to acquire such software and a lack of knowledge of data analytics to facilitate its use [3,4]. The realisation by SMEs of the benefits of timely decision making increases the demand for low-cost and user-friendly Cloud BI solutions as decision support systems [5–7]. Cloud BI provisioned as a Software-as-a-Service (SaaS) application is emerging as an alternative to the very complicated traditional BI which requires data analytics experts to use it [8,9]. SMEs that have perennial financial challenges in acquiring expensive traditional BI and hiring data analysts to use the complex applications can use Cloud BI as business solutions for data management and strategic decision making [3,4].

South African SMEs play an important economic role as they make up 95% of the businesses, employ close to 60% of the employees in all sectors, and contribute at least 40% to the gross domestic product [10]. The number of registered SMEs in the nine

South African provinces during the third quarter of 2019 was estimated to be close to 787,300 [11,12], as shown in Table 1.

**Table 1.** Distribution of small and medium enterprises (SMEs) in the 9 South African provinces.

| Province | Estimated of SMEs | % |
|---|---|---|
| Gauteng | 277,917 | 35.3 |
| KwaZulu-Natal | 154,311 | 19.6 |
| Western Cape | 124,393 | 15.8 |
| Mpumalanga | 61,409 | 7.8 |
| Limpopo | 48,025 | 6.1 |
| Eastern Cape | 41,727 | 5.3 |
| North West | 36,216 | 4.6 |
| Free State | 33,854 | 4.3 |
| Northern Cape | 9448 | 1.2 |
| Total | 787,300 | 100 |

The information in Table 1 shows that 71% of the SMEs are found mainly in the three most developed South African provinces, namely Gauteng (35.3%), KwaZulu-Natal (19.6%), and the Western Cape (15.8%), whereas the other six provinces account for less than 30%. Mpumalanga, Limpopo, the Eastern Cape, North West, and the Free State are mainly rural provinces, each consisting of a single administrative city and the rest rural towns with different commercial activities [11,12]. However, the COVID-19 pandemic has seriously affected most South African SMEs, especially in small towns which depend on manual and face-to-face business transactions [10].

Most SMEs face technological uncertainty as they are supposed to decide which cloud service to adopt and which cloud service providers (CSPs) to subscribe to amid increasing cybersecurity risks [2,4,11]. Patil and Chavan [7] suggest that the success of SMEs in migrating data from on-premises information systems to the cloud depends on the ability to select secure Cloud BI suitable for business needs as well as reliable CSPs. SMEs, particularly in small towns, face several challenges in this regard as their businesses are centred in areas where access to IT specialists is severely limited. This means that the important task of evaluating cloud services is left to the SME owners and managers who do not have the relevant IT expertise.

Regardless of many benefits that cloud services can offer, cybersecurity threats, privacy, and trust in and the reliability of CSPs are reported as the most influential factors likely to prevent SMEs from adopting and using Cloud BI applications [13,14]. In this context, the evaluation of cloud services and applications becomes an important undertaking for SMEs to adopt and use Cloud BI applications.

Currently, the few studies encouraging South African SMEs to adopt and use Cloud BI applications and other cloud services are silent on how these enterprises could evaluate and select the right solutions [3,11]. During the COVID-19 pandemic, to quickly adapt to the changing work environment and keep businesses going, SMEs had to make many compromises on using cloud services and other online applications to conduct business activities [15]. The literature highlights the dangers of indiscriminate adoption of cloud services without assessing issues related to security and privacy in the cloud environment [16,17]. Sceptics of cloud services adoption by SMEs allege that most of the existing security frameworks for evaluating Cloud BI are poorly aligned in terms of relevance and practicability, and therefore are not suitable for use by SMEs without the assistance of IT specialists [9]. Mirai Security [18] asserts that most of the existing security frameworks, industry-led guidelines, and standards are too massive and complex for SMEs to articulate and implement because they were designed for large enterprises with considerable IT infrastructure managed by specialists. Mirai Security [18] further posits that most of the time, enterprises are aware of the importance of cybersecurity but lack knowledge of where to begin and how to proceed using an existing framework to evaluate the security of cloud

services. Without appropriate assistance, SMEs have no option but to adopt cloud services without conducting a systematic evaluation process. Boonsiritomachai, McGrath, and Burgess [19] posit that SMEs require easy-to-use security evaluation methodologies or tools for BI to cater for their level of IT knowledge and skills, and to guide them in selecting appropriate business solutions. The perennial problem is that SMEs are expected to adopt and use new technologies that have only been tested to work for large enterprises. The few studies on Cloud BI and cloud services for South African SMEs do not adequately address the security evaluation of these important technologies. This suggests that SMEs adopt and use cloud services without due diligence, thereby putting these businesses at risk.

This study contributes to the existing knowledge by addressing the literature gap in Cloud BI security evaluation among SMEs. The adoption of Cloud BI application and other cloud services by SMEs in small South African towns, particularly in disadvantaged provinces, cannot be left to chance. To assist South African SMEs in small towns, it is important to have insight into (1) the security evaluation tools these enterprises use when selecting IT solutions; (2) the challenges they face when evaluating cloud services; and (3) the best practice aspects they consider to be important when selecting Cloud BI applications. The findings from the study and the best practices from existing standards and frameworks were used to conceptualise a security framework, which was then validated by information security specialists and SME decision-makers. Therefore, this study answered the following four questions:

RQ1: What security evaluation tools do SMEs use when selecting Cloud BI and other cloud services?

RQ2: What challenges do SMEs face when using existing security evaluation tools?

RQ3: What security evaluation best practice do SMEs consider when evaluating Cloud BI for adoption?

RQ4: What can be the main components of a security evaluation framework for Cloud BI applications suitable for SMEs in small towns?

The rest of the paper is organised as follows: South African economic sectors; security evaluation initiatives and frameworks; the research method; results; discussions of findings; and finally, the conclusions.

## 2. South African Economic Sectors

South Africa is one of the African countries whose economy is well developed and is supported by a strong economic infrastructure [12,20,21]. Due to its vast economy, close to 76% of the biggest African multinational companies have invested in various South African economic sectors [20]. The key sectors in the South African economy are agriculture, mining, manufacturing, wholesale and retail trade, financial services, transport, construction, tourism, and Information and Communication Technology (ICT) [12,21]. According to [20], very few South African SMEs participate in mining, agriculture, and manufacturing sectors which are generally dominated by large business enterprises (LBEs) but in the other sectors. All other economic sectors depend on the ICT sectors for ICT infrastructure and services [22]. The International Trade Administration [23] describes the South Africa ICT sector as one of the biggest, sophisticated, and fastest growing industry in Africa. The ICT industry provides technical leadership in Internet services, electronic banking services, the mobile software field, security software, and cloud computing services [12,23]. This makes South Africa a regional ICT centre and supply-base for all other countries in the region and the continent. Furthermore, South African companies partner with locally based international companies' subsidiaries to provide the bulk of fixed and wireless telecommunication networks within the country and across the continent [23]. Internet connectivity is available throughout South Africa, but the reliability depends on Internet service providers (ISP), the types of connectivity, and the infrastructure used by the enterprise. However, broadband is improving with the use of fibre optic, 4G and 5G technologies. A report on the General Household Survey by Stats SA (2018) [24] shows that at least 75% of the national household usage of the Internet is conducted over mobile devices due to the reduction in

the cost of data by the three major South African telecommunication companies Mobile Telecommunications and Networks (MTN), Telkom, and Vodacom.

Although several SMEs are key players in the ICT industry, either as service providers, software developers, or infrastructure retails, the use of ICT applications to support business operations is more prevalent with LBEs than SMEs, regardless of the locations of these enterprises. ICT infrastructure (hardware and software) is relatively expensive for SMEs [2]. The digital divide among South African communities has been explained by [25], in which cities have modern ICTs whilst small towns and rural areas have relatively old infrastructure which limits access to the Internet and web applications. This constrains business entities from participating in e-commerce and knowledge economy [12,23]. To address the ICT challenges faced by South Africans in small towns and rural communities, Internet satellite dishes and antennas have been installed [23,26]. In cities, e-commerce is well-established compared to small towns due to the reliable Internet service providers (ISPs), banking services, and connectedness with other institutions in various economic sectors. The use of online, web, and cloud applications by SMEs across South Africa has been on the rise particularly in cities and small towns [12]. According to [22], access to the Internet by the general public was highest in Gauteng (74.6%), Western Cape (72.4%), Mpumalanga (70.2%), and Eastern Cape (55.3%), and lowest in Limpopo (46.2%). This implies that the main access points for the Internet for many South Africans are public places such as educational facilities, Internet cafes, and workplaces [22,26] and some of these places are unsafe for online business transactions that require payments. Therefore, most of the South African consumers prefer traditional physical retail transactions due to fear of cybersecurity issues associated with online transactions [27]. With the COVID-19 pandemic continuing unabated, the use of online, web, and cloud applications is reported to be on the rise in the wholesale and retail, banking, and financial sectors [10,15].

The SA economy is vastly dependent on ICT services for data storage and processing, transactions, and communication and provides easy access to e-markets. This is true for both large and small enterprises; however, due to a digital divide, enterprises, particularly SMEs in small towns and rural areas, struggle in this area due to limitations in terms of ICT infrastructure because they tend to rely on on-premises facilities. Despite the challenges faced by SMEs in using ICTs, LBEs in South Africa are preparing for the fourth industrial revolution. This highlights a wide digital divide between SMEs and LBEs. This study focuses on Cloud BI applications.

## 3. Security Evaluation Initiatives for Cloud BI Applications

Many of the security evaluation strategies for Cloud BI applications used today are similar to those used in traditional information systems (ISs) because they depend on vulnerability assessment and penetration testing (VAPT) and the application of standards, frameworks, checklists, policies, and guidelines [28,29]. Regardless of their popularity with large enterprises, VAPT techniques are reported to be of little practical application in SMEs because of their small IT infrastructure and limited technical skills [17]. The conditions under which VAPT is performed, and the expertise required, reduce the possibilities of SMEs using these techniques to evaluate Cloud BI applications [30,31]. VAPT requires SMEs to utilise scanners to identify and analyse different security weaknesses in the enterprise IS devices and software owned by CSPs to ascertain the effectiveness of countermeasures [32]. VAPT is also reported to be effective with known vulnerabilities in technologies used, but not with financial risks that an enterprise can suffer due to the exploitation of the vulnerabilities by cyber threats [33,34].

Information security frameworks, standards, and methodologies have been developed to provide solutions to various IT problems faced by different enterprises [18]. These frameworks specify policies, controls, and procedures that enterprises should use to assess and measure the conformance of service providers to security standards [18,35]. Security frameworks and standards such as (a) the International Standards Organisation (ISO) 27,001, (b) Control Objectives for Information Technologies (COBIT), (c) National Institute

of Standards and Technology (NIST), (d) information security risk management (ISRM), and (e) enterprise risk management (ERM) have been used in LBEs by IT and security specialists. Security specialists are usually not found in SMEs where cloud computing technologies are still new [36,37]. Without technical assistance from IT security specialists, SMEs face challenges in using conventional frameworks when evaluating cloud services for adoption. Traditional frameworks and standards remain important because they address pertinent security issues that enterprises should consider during the adoption and use of traditional and cloud applications. However, the use of these security tools and techniques by SMEs in small rural South African towns when evaluating Cloud BI applications for adoption remains conjecture.

ISRM is a methodology used to identify, assess, and treat risks to the confidentiality, integrity, and availability of IT assets based on the overall risk tolerance of an enterprise [38]. The ISO 27,000 series identifies information security management system requirements in an enterprise [39]. An enterprise can address IT management and governance using COBIT [40]. Similarly, the NIST cybersecurity framework provides enterprises with standards and best practices to deal with interoperability, usability, and privacy in cloud services [18]. Lastly, ERM is used to identify, assess, and prepare an enterprise to prevent possible dangers, hazards, and events due to physical, natural, and human-made disasters to IT systems [41]. These traditional security frameworks and standards have extensive documentation requiring IT security expertise for successful implementation [37,42].

Complexities in industry frameworks and standards pose challenges to non-technical IT users such as SMEs that intend to evaluate and select Cloud BI for adoption and use for the first time. Challenges of using traditional security frameworks tend to lead to the proliferation of new and customised frameworks to address Cloud BI adoption and use [5,18]. Customised frameworks produce results quickly because they focus on specific aspects of the cloud technology associated with a particular group of enterprises, but the flaws of the original framework persist [43,44]. New security frameworks provide appropriate solutions to emerging adoption challenges that cannot be addressed by existing ones but are cumbersome to develop and validate [42,45].

SMEs need user-friendly security evaluation frameworks and techniques that can be used by decision-makers of these enterprises to assess and evaluate vulnerabilities, threats, and risks in Cloud BI before adoption [37]. Currently, there are no frameworks specific to Cloud BI for use by SMEs, especially for non-IT specialists in small South African towns. The European Union Agency for Network and Information Security [45] encourages enterprises to formulate their frameworks if existing ones do not meet their needs. However, developing a new framework is a laborious exercise for SMEs, and they are compelled to use existing ones selectively.

## 4. Methods

### 4.1. Population

In SMEs, decision-makers are responsible for the selection and use of various IT systems and, therefore, are in the best position to provide the information on security evaluation of Cloud BI. Similarly, IT security specialists know various cloud applications and are in a position to evaluate the proposed security framework. Therefore, data were collected from two samples: one of SME decision-makers and one of IT security specialists.

Limpopo is South Africa's northernmost province with a population of six million people [24]. It borders Mozambique, Zimbabwe, and Botswana, and this makes it strategically viable for trade and commerce [12]. The economy of the province is based on mining, farming, tourism, timber, ICTs, and wholesale and retail, both in the formal and informal sectors [11,24]. The number of towns in Limpopo is reported as 45. Polokwane is the only city in the province, and there are 10 mining towns, 7 farming towns, 21 rural towns, and 7 tourist towns. Of the 48,025 SMEs in Limpopo, 23% are found in Polokwane, 16% in mining towns, 5.6% in farming towns, 24.4% in rural towns, 10% in tourist towns, and 23% in rural villages across the province, participating in different economic sectors [12]. In this

study, the term "small town" refers to farming, mining, rural, and tourist towns [10,12]. However, there is no database or documentation of SMEs in individual small towns.

### 4.2. Sampling

The study sample consisted of decision-makers from SMEs already using various online, web-based, and cloud applications to support business operations in small towns. Five small towns were randomly sampled from the 45 in Limpopo. Due to the lack of a sample frame of SMEs using online business solutions and web applications in each of the small towns, the linear snowball sampling technique was used. Snowball sampling is a non-probability technique of selecting a survey sample primarily utilised to find rare or hard-to-locate populations by employing referrals or networks [35]. Several benefits of snowball sampling techniques have been documented. Sadoughi, Ali, and Erfannia [46] argue that snowballing techniques save time to obtain a representative sample for the study from the target population at a relatively low cost compared to other sampling techniques. Johnson [47] posits that snowball techniques are a low-cost and relatively efficient method for finding hard-to-find individuals. The use of the snowball technique in this study enabled the researcher to collect data from the respondents referred to by other respondents familiar with the population and online, web, and Cloud BI applications being studied.

For framework validation, a convenience sample of 35 IT security specialists was drawn from a population of IT personnel employed by various large private enterprises and municipality departments in the province. The sample from relevance acceptance validation was purposively selected from the SMEs who responded to the first questionnaire as well as IT specialists (lecturers) from the local university and two technical colleges.

### 4.3. Questionnaire Design, Validity, and Reliability

Three online survey questionnaires were designed and validated at different stages of this study. The online questionnaires were for collecting data from (1) SME decision-makers concerning security evaluation issues; (2) IT security specialists for framework review and relevance validation; and (3) SME decision-makers and other IT specialists for acceptance validation of the framework. The online questionnaire method has the advantage of collecting data from several respondents spread over a large geographical area at a reduced cost and within a short amount of time [48]. The design of the online questionnaires involved selecting and customising items from different sources to reduce potential common method variance (CMV) bias following [49]. The first question was designed to collect data about Cloud BI security evaluation issues from SME decision-makers. The second and third questionnaires were used to collect validation data from IT security specialists and decision-makers respectively. Questionnaire validity and reliability were catered for at the design stage by reducing CMV biases. According to [49], the bias in questionnaires arises from factors such as the poor design of individual questions and how the whole questionnaire is designed, administered, and completed. To improve the content and construct validity, questions used in questionnaires were adapted from previous studies on Cloud BI adoption conducted elsewhere in the world [45]. To ensure content and construct validity, the questionnaires were reviewed by four IT security specialists from private companies who advised the researchers on what to improve or omit. The questionnaire for decision-makers was piloted with five randomly selected SME owners, whilst that of IT security specialists was piloted with two IT lecturers from the University of Venda. Corrections such as editing and rearrangement of items were made. The reliability of each questionnaire was determined using SPSS Version 26, in which a Cronbach's alpha of 0.754 was obtained for the decision-makers questionnaire, 0.658 for relevance validation with IT security specialists, and 0.713 for acceptance validation with SME decision makers. According to [50], a Cronbach's alpha above 0.6 is acceptable and above 0.859 is a highly acceptable measure of reliability for a set of variables in a questionnaire. Cronbach's alpha

demonstrated that the reliability of each questionnaire was acceptable, showing consistency and links among variables used in each instrument.

*4.4. Data Collection*

To initiate network sampling and data collection with SMEs, the online questionnaire was sent to five purposively selected decision-makers of enterprises already using online IT systems. Each of the five decision-makers was asked to refer two other enterprises within their locality using IT systems. The researcher contacted the enterprises referred to and then requested the decision-makers to complete the online questionnaire. Sixty-eight SME decision makers were contacted, and the online questionnaire was emailed to them after obtaining their consent. Only 57 decision-makers completed the online questionnaire.

A five-component security evaluation framework together with a basic checklist for each component was conceptualised using results from primary data collected from SME decision-makers and secondary information on best practices provided by existing frameworks and standards. The proposed framework and checklists were then validated for relevance (content validation) by 35 IT security specialists, and for acceptance by 57 decision-makers already using various online, web, and cloud applications.

**5. Results**

SPSS was used to process and analyse data quantitatively. Results were presented using simple descriptive statistics such as frequency tables and graphs. The results were presented and interpreted in four subsections: (1) demographic information, (2) security evaluation tools used by SMEs, (3) challenges faced in using existing evaluation tools, and (4) best practice followed in evaluating Cloud BI before adoption.

*5.1. Demographic Information*

Demographic results provide useful information about the characteristics of the respondents in this study. Table 2 shows the distribution of types of SMEs surveyed across the five selected small towns in Limpopo.

**Table 2.** Distribution of surveyed SMEs by type.

| SME Type | Frequency | % |
|---|---|---|
| Car sales | 12 | 21.1 |
| Wholesales | 12 | 21.1 |
| Motor spares | 9 | 15.8 |
| Accommodation | 8 | 14 |
| Car rentals | 6 | 10.5 |
| Finance | 6 | 10.5 |
| Internet services | 4 | 7 |
| Total | 57 | 100 |

The results in Table 2 show that respondents were from SMEs participating in key economic activities, with car sales and wholesale at 21.1% each and motor spares at 15.8%. Internet services were provided by the least number of respondents. These results confirm that various SMEs were considering adopting and using Cloud BI and other cloud services.

Experience in using online and web applications to conduct business transactions and operations and support business decision-making gave insight into the main characteristics of the respondents of this study. Figure 1 shows the distribution of respondents by experience in using IT systems to support business operations.

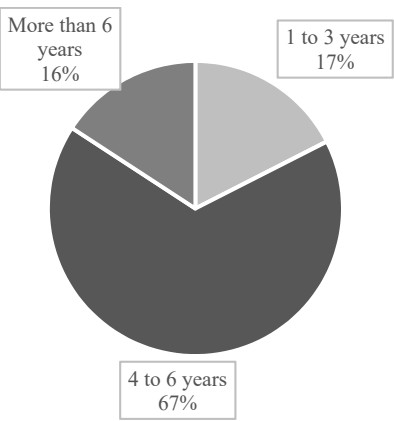

**Figure 1.** Experience in using online IT systems to support business operations.

Figure 1 shows that most of the respondents (82.5%) had been using IT systems to support business operations for at least four years. The results confirm that decision-makers had a good experience in IT use and this was crucial in answering further questions in the evaluation of Cloud BI applications.

This study sought information about the awareness of Cloud BI applications among decision-makers and the results are depicted in Figure 2. It can be seen that the awareness of Cloud BI applications among SME decision-makers from small South African towns was generally good, as indicated by 63.2% of the respondents.

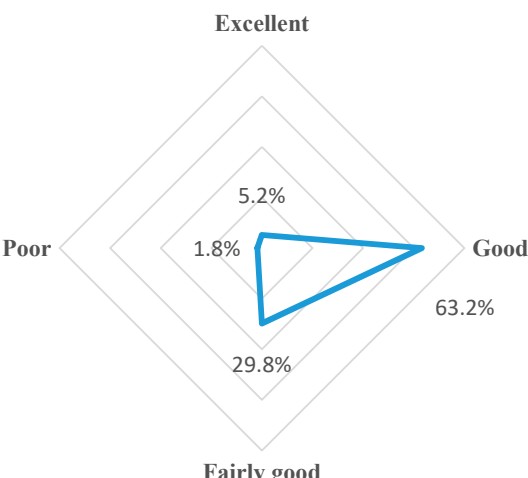

**Figure 2.** Awareness of Cloud BI applications.

It was important to ascertain the effort being made in the adoption of Cloud BI applications by SMEs and, therefore, the respondents were asked to indicate their current stage of adoption. Figure 3 depicts the distribution of respondents by their stage of Cloud BI adoption.

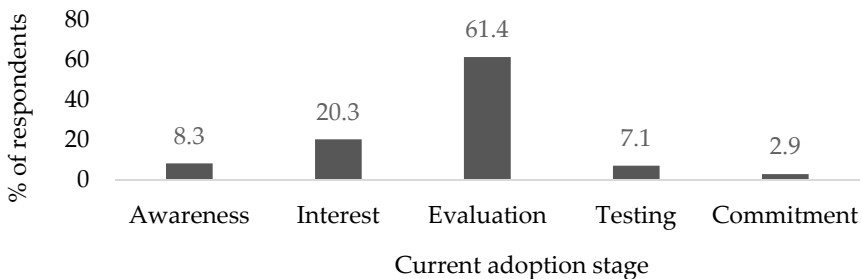

**Figure 3.** Stage of adoption of Cloud BI applications.

Only 2.9% of the respondents indicated that they had successfully adopted Cloud BI applications and most (61.4%) indicated that their enterprises were at the evaluation stage. The fact that most of the enterprises were at the evaluation stage means that decision-makers were facing challenges at this stage, causing delays in the selection of Cloud BI applications or even resulting in enterprises abandoning the idea of adopting applications.

*5.2. Security Evaluation Tools Used by SMEs in Selecting Cloud BI*

For insight into the evaluation tools used by SMEs, respondents were asked to indicate the tools they used when evaluating IT solutions in general. Figure 4 depicts the tools used to assess and evaluate Cloud BI applications and other IT solutions.

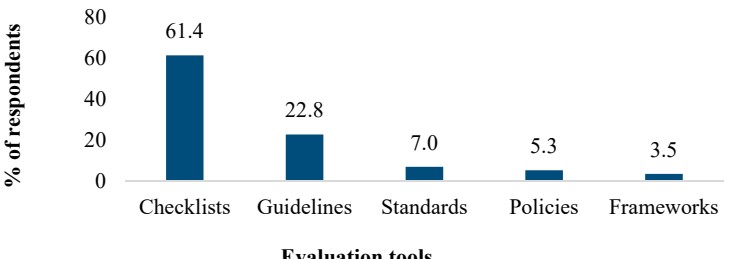

**Figure 4.** Tools used in evaluating Cloud BI and IT solutions.

Figure 4 shows that 61.4% of the respondents preferred to use checklists to evaluate Cloud BI applications, and 22.8% preferred guidelines over standards, policies, and frameworks (3.5%).

Respondents were further asked to indicate the industry security methodologies, standards, and frameworks referenced in the tools they used to evaluate Cloud BI applications. Figure 5 shows the results.

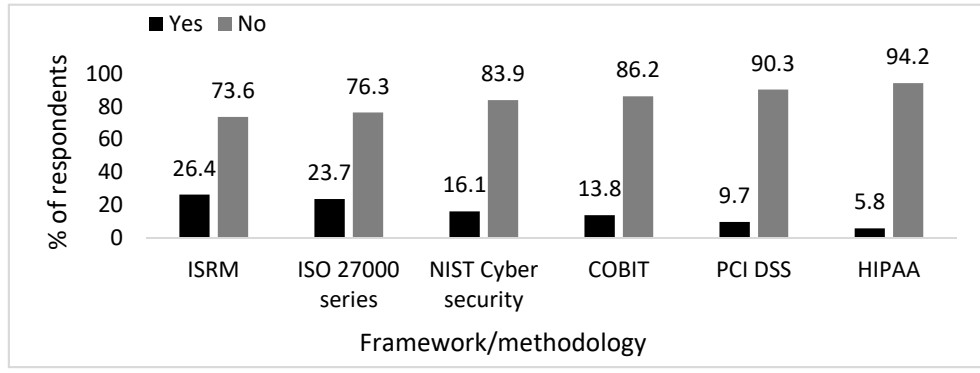

**Figure 5.** Frameworks and standards referenced in tools for Cloud BI evaluation.

Figure 5 shows that the majority of the respondents (73% to 94.2%) indicated that the evaluation tools they used were not based on industry security standards and frameworks. Between 13% and 27% of the respondents had used evaluation tools based on industry standards and frameworks such as ISRM, the ISO 27,000 series, and the NIST cybersecurity. However, less than 10% of the respondents had used PCI DSS and HIPAA tools. These results indicate that there are underlying issues that lead to the poor use of industry standards and frameworks among SMEs.

The study also investigated how decision-makers chose the evaluation tools they used for Cloud BI applications (Figure 6).

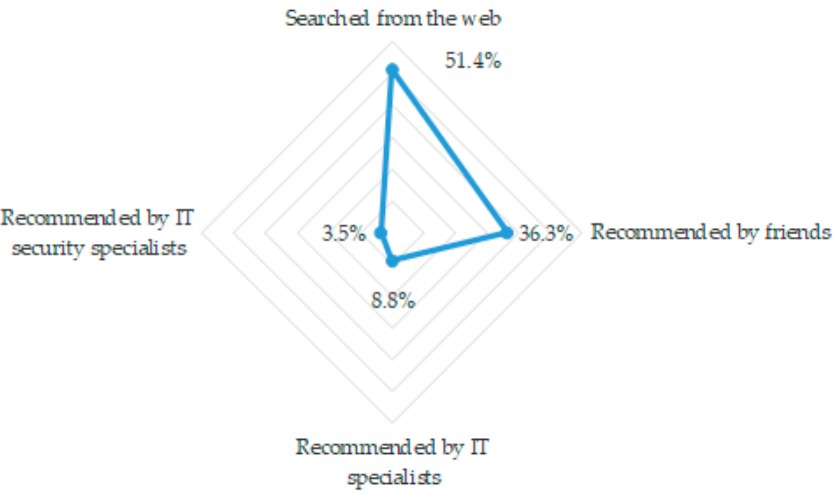

**Figure 6.** Choice of evaluation tools.

Figure 6 shows that the majority of the respondents (51.4%) searched for evaluation tools from the web and 36.3% used tools recommended by friends. Very few respondents received assistance from IT and security specialists. These results show that very few of the SMEs relied on IT and security specialists when selecting various online and cloud services.

*5.3. Challenges Faced by SMEs When Using Traditional Security Evaluation Tools*

To understand the challenges faced by SMEs in using existing evaluation tools, respondents were asked to rate a few items with constructs on the tools when selecting Cloud BI applications. A 3-point Likert scale (Agree, Not sure, and Disagree) was used. Table 3 shows that challenges faced by respondents were related mainly to human, technological, and time factors. At least 73% of the respondents affirmed that most of the challenges in using evaluation tools were due to human factors. Respondents stated that the lack of knowledge of tools to evaluate IT systems and limited skills of using existing tools were prevalent challenges to security evaluation in Cloud BI applications. They lacked confidence in using existing tools and were not able to receive assistance from IT specialists. These results show that SMEs require assistance in the types of tools to use and skills to use the tools. For technological factors, 85.9% of the respondents indicated that existing evaluation tools were very big and complicated to implement considering the lack of skills and knowledge of SME decision-makers. Similarly, 68.4% of the respondents stated that evaluation tools did not address problems that their SMEs faced in selecting IT systems. The majority of the respondents (88.8%) indicated that existing tools were not user-friendly, but were difficult to use when evaluating Cloud BI applications.

**Table 3.** Challenges faced in using existing security frameworks and standards.

| | Ratings (n = 57) | | |
|---|---|---|---|
| **Challenges in Using Evaluation Tools** | **Agree %** | **Not Sure (%)** | **Disagree (%)** |
| Human factors | | | |
| I do not know any tools used to evaluate IT systems | 73.4 | 2.1 | 24.5 |
| I have limited IT skills to use existing evaluation tools | 84.2 | 3.5 | 12.3 |
| I cannot choose a tool from several existing ones | 88.3 | 1.1 | 10.6 |
| I am unable to customise tools to suit enterprise needs | 75.4 | 3.5 | 21.1 |
| I am not confident in using evaluation tools | 90.8 | 2.5 | 6.7 |
| I cannot get proper assistance from IT specialists | 85.2 | 3.5 | 11.3 |
| Technological factors | | | |
| Evaluation tools are too big and complex for me to use | 82.8 | 5.1 | 12.1 |
| Tools do not address the problems faced by SMEs in selecting IT apps | 68.4 | 7.1 | 24.5 |
| I find evaluation tools user-friendly | 10.2 | 2.3 | 87.5 |
| I cannot customise existing security tools to meet business needs | 77.5 | 2.3 | 20.2 |
| Time factor | | | |
| I take a lot of time to understand evaluation tools | 85.9 | 3.5 | 10.6 |
| Existing tools require a lot of my time and effort to use | 74.8 | 4.3 | 20.9 |
| I take a lot of time to find a relevant evaluation tool from the web | 69.9 | 18.8 | 11.3 |

Finally, the time factor was regarded as a challenge by at least 74% of the respondents, who indicated that they needed more time and effort to learn how to use traditional evaluation tools. The SMEs had challenges in using tools for evaluating IT solutions, particularly Cloud BI. The majority of the respondents (84.2%) confirmed that the limited IT knowledge of SMEs made it difficult to use or customise the complex evaluation tools for the benefit of the enterprise. Consequently, the respondents perceived the evaluation tools as being inappropriate for use by SMEs. These results underline why SMEs do not make use of the security assessment tools, regardless of being aware of them. This implies that SMEs require easy-to-use frameworks commensurate with their IT knowledge and business niche. These results are an indication that SMEs could be using unsystematic strategies when selecting Cloud BI applications.

*5.4. Use of Existing Best Practices by SMEs When Selecting Cloud BI Applications*

The respondents were given a list of best practices that could be used when evaluating Cloud BI applications and were asked to indicate the likelihood of their using each if given a chance to do so. The respondents rated these on a 4-point Likert-type scale (Most likely, More likely, Less likely, and Not likely). Table 4 shows five categories of best practices from existing standards and frameworks that SMEs were prepared to use when evaluating Cloud BI applications.

5.4.1. Alignment of Data Management Processes and Security with Business Needs

For the three items under this category, 92% of the respondents indicated that they were more or most likely to undertake the activities when evaluating Cloud BI applications. The results confirm that data management processes and security were key to the business needs and should be aligned at the onset of the evaluation process.

**Table 4.** Likelihood of using existing best practices in evaluating Cloud BI applications for adoption.

| Best Practice Security Evaluation Statements | Ratings (n = 57) | | | |
|---|---|---|---|---|
| | Most Likely % | More Likely % | Less Likely % | Not Likely % |
| *Alignment of data management processes and security to business needs* | | | | |
| Identifying sensitive data to be migrated and managed in the cloud | 54.4 | 45.6 | 0 | 0 |
| Identifying enterprise data and application security needs | 35.1 | 57.8 | 7.1 | 0 |
| Deciding security requirements of data to be migrated | 36.8 | 59.6 | 3.6 | 0 |
| *Assessing Cloud BI operational and security functionalities* | | | | |
| Assessing the integration of Cloud BI applications with existing enterprise information system | 73.7 | 24.5 | 1.8 | 0 |
| Checking the usability of the application by standard users | 66.7 | 31.5 | 1.8 | 0 |
| Assessing access control and authentication features | 61.4 | 35 | 3.6 | 0 |
| Identifying Cloud BI data management functionalities | 64.9 | 33.3 | 1.8 | 0 |
| Evaluating the effectiveness of security controls in each Cloud BI application | 45.6 | 50.8 | 3.6 | 0 |
| Identifying security vulnerabilities, threats and risks | 45.6 | 49.1 | 5.3 | 0 |
| *Assessing Cloud deployment models security vulnerabilities, threats, and risks* | | | | |
| Assessing data portability and cloud interoperability | 80.7 | 19.3 | 0 | 0 |
| Assessing data accessibility publicly by unauthorised cloud users | 68.5 | 31.5 | 0 | 0 |
| Verifying security breaches by CSP employees and other cloud tenants | 66.6 | 22.8 | 10.6 | 0 |
| *Assessing security, trust and reliability of Cloud service providers* | | | | |
| Assessing service reliability and performance of CSPs | 84.3 | 15.7 | 0 | 0 |
| Checking security responsibilities of enterprises and CSPs | 72.8 | 28.2 | 0 | 0 |
| Assessing security reliability of CSPs | 78.9 | 19.3 | 1.8 | 0 |
| Scrutinising the terms and conditions of contracts and service level agreements (SLAs) | 68.3 | 28.1 | 3.6 | 0 |
| Requesting reports on service downtime and unavailability to users | 57.8 | 36.9 | 5.3 | 0 |
| Assessing CSP's adherence to certification and standards | 56.1 | 42.2 | 1.7 | 0 |
| Assessing CSP's data security and governance | 54.4 | 43.8 | 1.8 | 0 |
| Confirming the level of control of data in the cloud of the enterprise | 42.2 | 54.2 | 2.5 | 1.1 |
| Scrutinising the business standing of CSP history | 28.1 | 56.3 | 10.4 | 5.2 |
| Assessing the physical security of the CSPS | 24.6 | 40.3 | 22.8 | 12.3 |
| *Assessing potential financial risks of using Cloud BI applications* | | | | |
| Identify loss of revenue due to downtime of the cloud service | 85.9 | 14.1 | 0 | 0 |
| Assessing the cost of subscribing to Cloud BI applications | 82.5 | 12.2 | 5.3 | 0 |
| Assessing litigation costs by customers after exposure of sensitive data | 64.8 | 31.6 | 3.6 | 0 |
| Assessing financial risks due to hidden subscription costs | 56.1 | 42.1 | 1.8 | 0 |
| Assessing penalty cost for misuse of service | 22.8 | 61.5 | 12.2 | 3.5 |

5.4.2. Assessing Cloud Models' Security Vulnerabilities, Threats and Risks

The majority of the respondents (66% to 80%) indicated that they were most likely to assess security vulnerabilities and cyber threats and risks of cloud deployment models based on three items in this category: data security, cloud portability, and interoperability. Respondents also considered assessing data accessibility by unauthorised cloud users in the public cloud as a best practice for SMEs when selecting Cloud BI applications. This is meant to verify types of security breaches by CSP employees and other cloud tenants. This shows that SMEs were aware that different cloud deployments could suffer security breaches due to different vulnerabilities.

5.4.3. Assessing Security, Trust and Reliability of CSPs

The results show that the majority of the respondents (54% to 84%) were most likely to consider seven of the ten activities when evaluating Cloud BI applications. Assessing security, service reliability, and performance of CSPs and responsibilities was considered important in the selection of Cloud BI by SMEs. Data security and service reliability provided by CSPs to clients were rated as being central to the assessment of the providers. The ratings show that trust in service provision, data governance, and adherence to security standards should be prioritised when selecting CSPs. Respondents also indicated that they were more likely to consider the business standing of the CSP and physical security.

5.4.4. Assessing the Potential Financial Risks of Using Cloud BI

Most of the respondents indicated that they were most likely to assess financial risks due to cybersecurity risks in the cloud and using Cloud BI applications. At least 84.5% of the respondents indicated that financial risks were important to consider when evaluating Cloud BI. Most of the respondents (84.3%) felt that SMEs were most likely to assess losses due to downtime of services; 82.5% were worried about the cost of subscribing for Cloud BI and litigation costs. As a means to avert such losses, there was a need to assess financial risks based on the suggested items.

*5.5. Components of the Conceptual Framework, Checklists and Validation*

5.5.1. Components of the Conceptual Framework

For the major components of the security evaluation framework, respondents from SMEs and IT and security specialists were asked to indicate which category of best practices were suitable for inclusion. The results are shown in Table 5. The majority of respondents (77% to 89%) indicated that the major categories of the best practices could be used as components or pillars of the framework. Similarly, the majority of the IT and security specialists (85–97.1%) affirmed that for a simple framework, the categories of the best practices were suitable to be used as the major components of the framework.

**Table 5.** Suitability of best practice categories as components of a security evaluation framework.

| Proposed Component of the Framework | SME Respondents (n = 57) | | IT Security Specialist (n = 35) | |
|---|---|---|---|---|
| | **Yes** | **No** | **Yes** | **No** |
| Aligning data management processes and security with business needs | 80.7 | 19.3 | 94.2 | 5.8 |
| Cloud BI operational and security functionalities assessment | 84.2 | 15.8 | 91.4 | 8.6 |
| Cloud deployment security vulnerabilities, threats and risk assessment | 77.2 | 22.8 | 82.9 | 17.1 |
| Security, trust and reliability assessment of CSPs | 89.5 | 10.5 | 97.1 | 2.9 |
| Financial risks of using Cloud BI applications assessment | 82.5 | 17.5 | 85.7 | 14.3 |

Based on these findings, a five-component security evaluation framework for Cloud BI was conceptualised comprising: (1) enterprise data and application security needs, (2) Cloud BI operational functionalities and security features, (3) cloud deployment models, (4) security, trust, and reliability of CSPs, and (5) financial risks of using Cloud BI. Specific

items to focus on during evaluation were included for each aspect and these are to assist in implementing security standards and frameworks. The framework was validated independently for relevance by 35 IT security specialists, and acceptance by 57 SME owners. Figure 7 depicts the conceptualised security evaluation framework for Cloud BI by SMEs. The five components are linked to show the composition of the framework. The framework starts with assessing the alignment of data management and security with business need and culminates in financial risk assessment, the main concern among SMEs arising from their limited financial resources. SMEs' financial requirements bring about challenges when developing a framework to meet such needs, hence the uniqueness of the conceptualised framework.

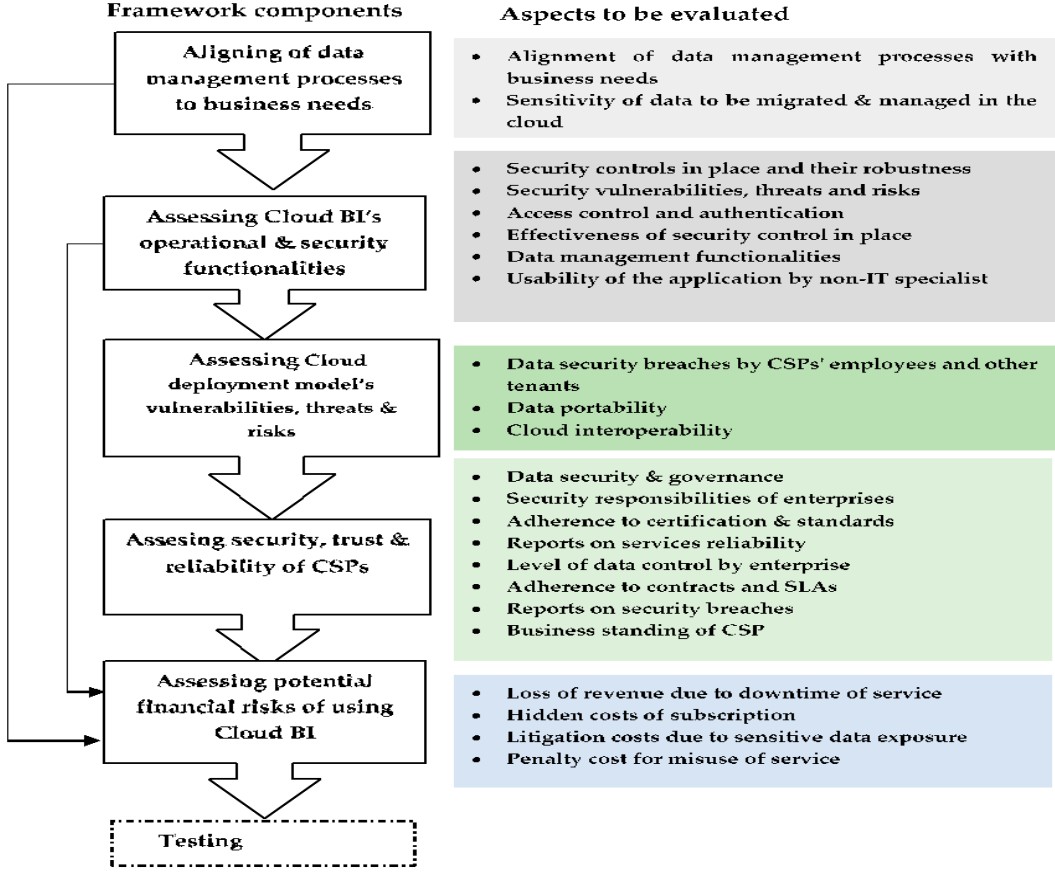

**Figure 7.** A conceptualised security evaluation framework for Cloud BI by SMEs.

### 5.5.2. Checklists for Use with the Framework

A checklist for use by decision-makers was created for each of the five components to be evaluated. Each checklist has several evaluation criteria and statements to guide the user on what to check during the data gathering process. A score of 1 is given for every true criterion; otherwise, 0 is given. The Actual score is the number of 1s on each checklist and this is compared with the *Expected score*. At the end of each stage, the user counts the number of 1s and records the actual score. An acceptable *Actual score* of 90% of the *Expected score* is needed for sensitive data and 70% for non-sensitive data. To move from the first to the second component, the former should be at least 85% of the enterprise's business needs (the *Actual score* should be 85% of the *Expected score*). The evaluation checklists are shown in Figures 8–13. The description of how to use the checklists during the evaluation is given after the figures.

| 1. ASSESSING BUSINESS NEEDS AND DATA SECURITY REQUIREMENTS IN THE CLOUD |||
|---|---|---|
| **Date of assessment:**……………………………………….…….. |||
| **Instructions:** These statements represent a checklist of key characteristics to consider when assessing the alignment of business needs and data security requirements. Please indicate in the score column whether or not the information provided in the statement meets each criterion for each statement *(1 = criterion met; 0 = criterion not met)* |||
| **Evaluation criteria** | **Statement description** | **Score** |
|  | Most of the enterprise data is in electronic format and processed electronically |  |
| **Alignment of data management systems with business needs** | Enterprise data is updated regularly |  |
|  | Enterprise data management plans support business plans and needs |  |
|  | The enterprise is already using cloud services for data storage |  |
|  | Storage is a reason to migrate data to the cloud |  |
| **Classification data to be migrated to and managed on the cloud regarding sensitivity or security needs** | Sensitive data is (can be) segregated from non-sensitive data |  |
|  | Data is (will be) accessible only to authorised users of the system |  |
|  | I can identify and decide on which data to migrate to the clouds |  |
|  | Data security is a reason for migrating sensitive data to clouds |  |
| **Security requirements of data to be migrated to and managed on the cloud** | Enterprise information security program is supported by policies, procedures, standards, regulatory and compliance requirements |  |
|  | Data in the cloud will be protected by — Passwords |  |
|  | Data in the cloud will be protected by — Encryption |  |
|  | In your enterprise, a migration roadmap directs data classification and protection |  |
|  | Enterprise data requires a high level of security to be stored in the cloud |  |
|  | Enterprise data security goals are easy to understand and simple to implement |  |
|  | Only authorised persons will access and manage data in the cloud |  |

| | |
|---|---|
| **Expected score** | 17 |
| **Actual score** |  |
| **% of Actual score/ Expected score** |  |

*(if % < 85%: align data management processes and security requirements to enterprise business needs)*

**Figure 8.** Checklist for assessing business needs and enterprise data security requirements.

| 2. ASSESSING CLOUD BUSINESS INTELLIGENCE APPLICATION USABILITY |||
|---|---|---|
| **Name of Cloud BI:**…………………………………….. **Date of Assessment:**…………………………………. |||
| **Instructions:** These statements represent a checklist of key characteristics to consider when evaluating a cloud business intelligence. Please indicate in the score/answer column whether or not the information provided in the statement meets each criterion for each statement *(1 = criterion met; 0 = criterion not met)* |||
| **Evaluation criteria** | **Statement description** | **Score** |
| **Functionalities of Cloud BI on key data management and security** | The Cloud BI app meets enterprise data management needs |  |
|  | Most features of Cloud BI app support current enterprise operations |  |
|  | The Cloud BI app connects to and supports on-premises data sources or systems |  |
|  | The Cloud BI app can connect to popular cloud data warehouses and databases |  |
|  | Users interact with the Cloud BI app mainly through the web browsers |  |
|  | Cloud BI app can run existing on-premises hardware |  |
|  | Reviews by other customers prove the claims by CSP what the Cloud BI app can deliver |  |
|  | The Cloud BI app is built mainly for all different business users |  |
|  | Cloud BI can easily be used by non-technical end-users |  |
| **Security vulnerabilities, threats and risks in shortlisted Cloud BI app** | The Cloud BI app features that support enterprise data sources, filters, data visualisations are easy to understand |  |
|  | The Cloud BI app has security features to protect data during processing and transit |  |
|  | The interface of the Cloud BI is safe, simple and easy to use by non-technical users |  |
|  | Cloud BI app interface does not remember access details, particularly passwords |  |
|  | Data files from on-premises are compatible with Cloud BI app and vice versa |  |
| **Security controls in place and their robustness** | User interface provides strong access control to the Cloud BI app |  |
|  | Cloud BI app provides for authorisation to authorised users only |  |
|  | Cloud BI app sessions time-out and automatically terminate all connections |  |
|  | Only the administrator can grant access privileges to the Cloud BI app |  |
| **Cloud BI's usability by none technical users** | The Cloud BI app can be acquired, run, used and audited with much ease |  |
|  | It is easy to learn how to use the Cloud BI app without formal training |  |
|  | Online tutorials and YouTube videos on Cloud BI app are available to assist users |  |
|  | Free training is provided for the Cloud BI app |  |
|  | The learning or training material is relevant to the enterprise needs |  |
|  | Online experts are available to provide needed support when a client faces challenges |  |
|  | The Cloud BI app is supported by a strong and reliable online community, forums, enthusiast blogs, passionate users or user group |  |

| | |
|---|---|
| **Expected score** | 25 |
| **Actual score** |  |
| **% of Actual score/ Expected score** |  |

*(if % < 85%: repeat this process with another CBI until you find one that closely meets your requirements)*

**Figure 9.** Checklist for assessing Cloud BI's operational and security usability.

**3. CLOUD DEPLOYMENT MODELS ASSESSMENT**

Type of deployment model:………………………………………….……

Date of assessment:……………………………………………………..

Instructions: These statements represent a checklist of key characteristics to consider when evaluating a cloud deployment. Please indicate in the score/answer column whether or not the information provided in the statement meets each criterion for each statement *(1 = criterion met; 0 = criterion not met)*

| Evaluation criteria | Statement description | Score |
|---|---|---|
| Security vulnerabilities, threats and risks in the cloud | Cloud allows Cloud BI app to easily integrate with on-premises systems | |
| | Vulnerabilities, threats and risks on on-premises systems are known | |
| | Vulnerabilities in the cloud models are already known or easy to locate | |
| | Threats and risks to the data and application in the cloud are easy to manage | |
| | Any reported security breaches reported for this model | |
| | Data segregation among users of different enterprises is provided | |
| Effectiveness of security controls | Security controls to identified threats and risks are in place | |
| | Security controls are effective against identified threats | |
| | The cloud deployment model does not disrupt enterprise operations | |
| | Security controls updated regularly in line with security regulations | |
| | The cloud encrypts all data at rest in databases, file systems and VM layer | |
| Availability, reliability and performance of the cloud | Cloud services are always available most of the time whenever required | |
| | Security in the cloud model is reliable in preventing data breaches | |
| | Performance of the cloud model meets the enterprise expectations | |
| Cloud interoperability and application portability | Cloud is interoperable with other clouds and on-premises systems | |
| | Cloud allows applications and data to move to other clouds or on-premises systems | |
| | Data files from the cloud can be accessed easily by on-premises applications without conversion | |

| | | |
|---|---|---|
| (Set own % of security requirement and assess each cloud deployment model to your expectation. % should be 75 to 95%) | **Expected score** | **17** |
| | **Actual score** | |
| | **% of Actual score/ Expected score** | |

**Figure 10.** Checklist for selecting cloud deployment model.

**4. ASSESSMENT OF CLOUD SERVICE PROVIDERS**

Name of CSP: ……………………………………………………….….. Date of assessment:……………………

Instructions: These statements represent a checklist of key characteristics to consider when evaluating a cloud service provider. Please indicate in the score/answer column whether or not the information provided in the statement meets each criterion for each statement *(1= criterion met; 0 = criterion not met)*

| Evaluation criteria | Statement description | Score |
|---|---|---|
| Data governance, accountability and security | CSP implements and adheres to security policies to protect clients from threats and data losses | |
| | CSP maintains integrity by data encryption to prevent unauthorised access | |
| | CSP preserves data confidentiality with strong authentication to prevent unauthorised access | |
| | CSP uses backup and recovery schemes to maintain availability | |
| | CSP offers reliable security controls to protect applications and data | |
| | CSP prevents security breaches by denying anonymous users access to services | |
| Internal security control through certification and standards for compliance | CSP has proof of compliance and audit performed regularly | |
| | CSP has proof of data centre protection that enterprises can assess | |
| | CSP keeps a record of end-user log activities as proof of security assurance | |
| | The security standards claimed by CSP are in place and adhered to | |
| | Areas of responsibilities shared between CSP and customer are clearly stated | |
| | CSP can assist the enterprise in meeting compliance standards that apply to industry | |
| | CSP clearly states enterprise responsibilities and support to be given | |
| Trust, reliability and history performance | CSP usually alerts clients of security breaches in the service in time | |
| | CSP provides clients with proof that penetration testing is done regularly | |
| | CSP has robust online communities who depend on the provider's fame | |
| | CSP is readily available to give necessary support to clients who face challenges | |
| | CSP is capable of overcoming most security issues faced by clients who use the services | |
| | The CSP is clear on the type of technical support being given (paid or free) | |
| | CSP's technical support is in line with the expectations of the enterprise | |
| Business continuity | Background checks on possible fraudulent business activities by CSP have been made | |
| | Enterprise data will be safe from security breaches with this CSP | |
| | CSP is capable of attending to system disruptions/outages | |
| | CSP can detect those who engage in malicious or fraudulent activities in the cloud | |
| | CSP has clearly stated or specified intellectual property ownership | |
| | There are clear terms for account termination with the CSP | |
| Cloud service agreement (AUP, SLA and contracts) | CSP reserves the right to share customer data with third parties and are conditions well stated | |
| | Assess the trust of cloud provider in doing the right thing using a legal agreement that will back up the enterprise if something goes wrong | |
| | CSP provides customers with clear information on controls, security and operation of the service on CSA, AUP & SLA | |
| | Data ownership, shared responsibilities, non-disclosure agreements, dispute handling are laid out clearly and easy to understand | |
| | SLAs cover essential requirements in terms of availability, response time, capacity and support | |
| | All legal requirements for the security of data hosted by CSP are clearly stated and enforceable by the parties involved | |

| | | |
|---|---|---|
| (Set own % between 75 to 95%) CSP security trust, reliability, performance and data governance. Your CSP can be the provider of Cloud BI or host) | **Expected score** | **32** |
| | **Actual score** | |
| | **% of Actual score/ Expected score** | |

**Figure 11.** Checklist for assessing CSP security, reliability, and performance.

| 5. ASSESSMENT OF FINANCIAL RISKS | | |
|---|---|---|
| Name of Cloud BI:.................................................: Date of assessment:........................................... | | |
| Instructions: These statements represent a checklist of key characteristics to consider when evaluating financial risks. Please indicate in the score/answer column whether or not the information provided in the statement meets each criterion for each statement *(1 = criterion met; 0 = criterion not met)* | | |
| **Evaluation criteria** | **Statement description** | **Score** |
| **Financial benefits of Cloud BI** | Cost of Cloud BI are substantially lower than acquiring traditional BI | |
| | Costs of training users of Cloud BI apps are lower than traditional BI | |
| | Cost of supporting and maintaining Cloud BI is lower than traditional BI | |
| **Assess hidden costs** | Initial subscription is clearly stated and fixed | |
| | An itemised bill for agreed services can be obtained from CSP | |
| | Notices of increase in billing are given prior | |
| | Clients always alerted of unutilised billable resources | |
| | Is CSP technical support for free | |
| | Expenses for data migration are included | |
| **Costs due to downtime** | Only uptime is paid for | |
| | Frequency of downtime does not affect business operations financially | |
| | CSP compensates enterprise for business loss due to outages | |
| **Litigation costs** | CSP is fully liable for litigations for data breaches on behalf of clients | |
| | CSP compensates customers for poor services on behalf of clients | |
| | SLAs have no disclaimers for unauthorised data access and hacking | |
| **Penalty costs** | CSP charges fines for any misuse of resources | |
| | Penalties resulting from non-compliance and not turning off some services are stated | |
| **Cost of Cloud BI app** | The Cloud BI app is free of charge or at an affordable cost to the enterprise | |
| | A free trial version is provided for users | |
| | The app easily configured by non-technical users | |

| (Set own % between 90 to 98% financial safety (2 to 8 % risk acceptance). If the financial risk > 15% repeat the steps 2 to 5 with other Cloud BI app, and providers) | **Expected score** | **20** |
|---|---|---|
| | **Actual score** | |
| | **% of Actual score/ Expected score** | |

**Figure 12.** Checklist for assessing financial risks associated with Cloud BI applications.

| Cloud BI Name:.................... | | | | | | |
|---|---|---|---|---|---|---|
| **COMPONENT ASSESSED** | **Expected Score** | **Actual Score** | **Actual %** | **Minimum expected %** | **Decision** | |
| 1 | Business needs and data security requirements in the cloud | 17 | | | | Accept | Reject |
| 2 | Cloud business intelligence application usability | 25 | | | | Accept | Reject |
| 3 | Cloud deployment models | 17 | | | | Accept | Reject |
| 4 | Cloud service provider's trust, reliability and performance | 32 | | | | Accept | Reject |
| 5 | Assessment of financial risks | 20 | | | | Accept | Reject |
| NB: *Final decision: Accept if all accept; reject for at least one component with a reject decision* | | | | | ***Accept*** | ***Reject*** |

**Figure 13.** A summary sheet for each set of evaluations.

To start the evaluation process, Checklist 1 should be completed to determine if an enterprise's data management processes and security requirements are or need to be aligned with the business needs. This is important in determining the readiness to adopt Cloud BI application and to migrate data. The other four checklists are completed sequentially depending on when the minimum actual percent for the current component is met. Each complete evaluation is recorded on a summary sheet shown in Figure 13. The actual percentage and minimum expected percentage for each component are compared. If the actual percentage is greater than or equal to the minimum expected percentage, the component is provisionally accepted for the Cloud BI; otherwise, it should be rejected. The Cloud BI application is accepted if the decision for all components is *Accept*.

Figure 13 shows a summary sheet which can be used to collate the results for each set of evaluations and to make decisions. Decision-makers have to complete one for each set of evaluations made.

### 5.5.3. Framework and Checklist Validation

The conceptualised framework and checklists were validated for relevance and acceptance using content and face validation techniques, respectively. Rykiel [51] regards validation as the process of evaluating a system or its component during or at the end of the development process to determine whether it is acceptable for its intended use and that it meets specified performance requirements. Content validation was also used as a form of relevance validation for the conceptualised framework as recommended by Krippendorff [49]. Studies show that content validation can be used in assessing the relevance and applicability of the framework to the context of enterprises used [49,52]. Content validation involves preparing a content validation form, selecting a review panel of experts, conducting content validation, reviewing the domain and items, providing a score for each item, and calculating the content validation index [53]. In this study, the content validation process focused on establishing whether the components and checklist items were representative of the evaluation process to be performed by the security framework when used by decision-makers [54]. The basic relevance validation involved collecting data from 35 IT security specialists working independently and from established IT organisations.

The framework and checklists were sent to the validators together with validation instructions. Two weeks were given for the reviewers to go through the framework and checklists. All 35 IT security specialists completed the online relevance (content) validation questionnaire. The 57 SME decision-makers involved in the initial data collection process completed the online acceptance validation (verification) questionnaire. A simple Pearson correlation relational analysis was conducted to determine the association of variables and the level of acceptance indicated by respondents. The results are shown in Table 6.

**Table 6.** Correlation results for relevance validation.

| Component of the Security Framework | The overall Relevance of the Framework | |
| --- | --- | --- |
| | Pearson Correlation | Sig. (2-Tailed) |
| Aligning data management processes and security with business needs | 0.735 ** | 0.000 |
| Cloud BI operational and security functionalities assessment | 0.407 ** | 0.004 |
| Cloud deployment security vulnerabilities, threats and risk assessment | 0.782 ** | 0.000 |
| Security, trust and reliability assessment of CSP | 0.671 ** | 0.002 |
| Financial risks of using Cloud BI applications assessment | 0.674 ** | 0.002 |
| The relevance of framework in addressing traditional security standards and frameworks during the evaluation | 0.509 ** | 0.020 |
| The relevance of checklist criteria in addressing the evaluation of key issues in Cloud BI adoption | 0.601 | 0.021 |
| Evaluation summary sheet | 0.642 | 0.011 |

** Correlation is significant at the 0.01 level (2-tailed).

The results show a significant association between the overall relevance of components and suggested activities of the Cloud BI security evaluation framework and the overall score of all measured aspects at $p < 0.05$. These results show that the conceptualised processes and activities of the formulated framework were regarded as relevant in the evaluation of Cloud BI by non-IT specialists. The acceptability of the framework was based on the overall ratings of each major component using a 3-item Likert-type scale (Highly acceptable, Acceptable, and Not acceptable). The results are presented in Figure 14.

The results show that all five components of the framework were rated as highly acceptable by at least 52% of the reviewers. This confirms that all the activities in each component were validated as being acceptable to be included in the evaluation framework for use by non-IT specialist decision-makers in SMEs in the five towns.

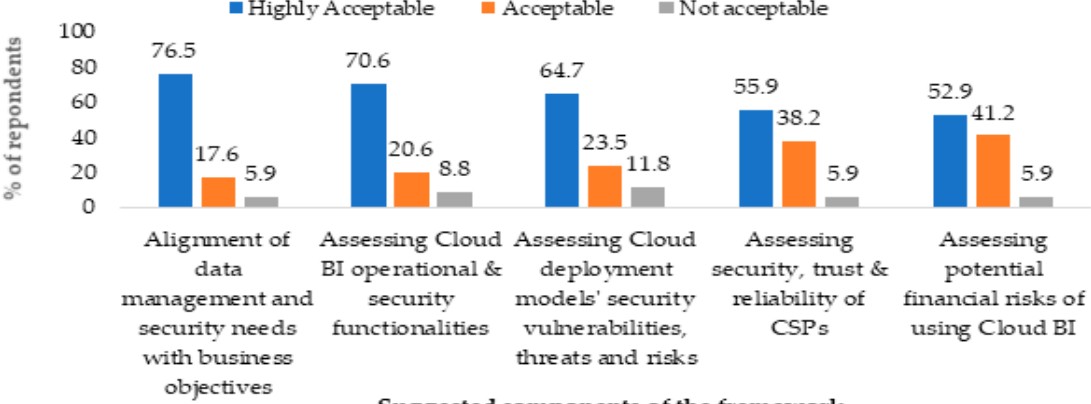

**Figure 14.** Acceptance validation results.

## 6. Discussion of Findings

The findings of this study are discussed based on three research questions.

### 6.1. RQ1: What Security Evaluation Tools Do SMEs Use When Selecting Cloud BI and Other Cloud Services?

The study found that checklists and guidelines were commonly used by the SMEs in evaluating IT solutions and these could be extended to Cloud BI. However, standards and frameworks were seldom used by the SMEs. There is a plethora of literature on the use of checklists as evaluation tools and in conjunction with other tools in IT systems. A study by Müller and Rood [55] reports that checklists are among the evaluation tools widely used by many organisations. There are many advantages of using checklists in security evaluation [56]. Unlike automated VAPT, an evaluation checklist can easily be used by potential cloud clients to compare cloud services from various CSPs in terms of security and other criteria [56,57].

Guidelines are recommendations, best practice, or support documents and processes that help with the interpretation and implementation of a specific policy or requirement [58]. This implies that guidelines can be of help to SMEs with some direction of implementing policies. For example, the NIST guidelines address vital issues in cloud security and privacy, namely architecture, identity and access management, trust, software isolation, data protection, compliance, availability, and incident response [59]. The main problem with most of the security initiatives by the NIST is that they can only be implemented by security specialists, which are personnel not found in SMEs. The study found that frameworks were the least used among SMEs due to challenges faced by non-IT specialists in these enterprises.

### 6.2. RQ2: What Challenges Do SMEs Face When Using Existing Security Evaluation Tools?

This study found that poor usage of standards, procedures, and frameworks among SMEs was due to limited knowledge about the tools as well as the complexities they pose to non-IT specialist users. Existing studies show that limited knowledge of cloud security and the inability to use existing conventional frameworks and techniques to evaluate cloud solutions remain major challenges among SMEs [59,60]. Furthermore, these security tools are suitable for mitigating cyber threats in large enterprise IT environments [18]. This implies that SMEs need user-friendly frameworks tailored to them to apply security standards and policy when evaluating cloud services. Several studies on the adoption of cloud computing among SMEs overlook challenges faced during the evaluation of cloud technologies [35]. Consequently, limited skill and knowledge in using existing evaluation tools remain one of the major challenges faced by SMEs in selecting Cloud BI and other cloud services. The implementation of conventional security frameworks requires IT

specialists. To encourage SMEs to evaluate cloud services before adoption, a user-friendly framework within their knowledge and security abilities is needed.

*6.3. RQ3: What Security Evaluation Best Practice Do SMEs Consider When Evaluating Cloud BI for Adoption?*

The findings discussed in this section entail the results in Table 4. The study found that despite the lack of knowledge of security evaluation issues, tools used, and the best practices from existing frameworks, respondents were eager to implement suggested methodologies as long as they work. According to Mirai Security [18], SMEs need to be aware of the frameworks and standards as the best practices they provide so that they can select what works for the enterprise. Mirai Security [18] also reports that SMEs are eager to put into practice what is best for them to use in supporting their business enterprises. However, the authors in [37] point out that the enterprises need a great deal of assistance to understand basic cloud computing and the services provided. For this current study, decision-makers indicated that most of the customised best practices provided were easy to understand and could be easy to implement when evaluating Cloud BI applications. The majority of the respondents (75% to 87%) affirmed that they were prepared to implement best practices for assessing the potential financial risks of using Cloud BI applications and those to do with security, trust, and reliability of CSPs. Independent studies confirm that SMEs require assistance for them to adopt and use existing standards and frameworks [37,61]. However, existing standards and frameworks provide vast and complex best practices that SMEs and government organisations might find difficult to apply when selecting Cloud BIs [45]. Due to these complexities, SMEs face a dilemma in articulating industry standards and frameworks [46] and this makes them hesitant to make mistakes by adopting the wrong technologies. The literature suggests that SMEs should be assisted with baseline guidelines on how to adapt and use cloud services [38,45,46]. Concerning SMEs in Limpopo, the decision-makers indicated their interest in adopting Cloud BI applications when they are assisted in evaluating and selecting the appropriate applications. What is encouraging is that a high level of awareness of Cloud BI applications and eagerness to evaluate the technology by decision-makers provides a strong basis for assisting these enterprises.

*6.4. RQ4: What Can Be the Main Components of a Security Evaluation Framework for Cloud BI Applications Suitable for SMEs in Small Towns?*

The framework was conceptualised using mainly the best practices and the input from decision-makers and the IT and security specialists who completed the questionnaire.

6.4.1. Aligning Business and Data Security Needs

Based on the given findings, the different building blocks of the proposed security evaluation framework will be discussed. The framework suggests that at the onset of the evaluation process, SMEs should assess business and data security needs. This consists of three vital information security risk management aspects to align their data management processes with business needs. To achieve this, SMEs will have to identify all the data and applications that should be migrated to the cloud and then classify the data according to their sensitivity. This will enable SMEs to decide on appropriate security requirements of data to be used as criteria for the security evaluation of the Cloud BI's operational and security functionalities. By doing this, SMEs will be able to independently implement the ISRM, ERM, or COBIT framework that requires enterprises using IT systems to be systematic in security evaluation [18,62]. Cloud BI depends on data stored in many online storage systems; data security and protection, therefore, play an important role in maintaining confidentiality, integrity, and availability [29]. Consequently, risk analysis of existing information systems is crucial and should be performed at the beginning of the evaluation process. The security framework considers important aspects of data security and protection for the successful adoption of Cloud BI applications by SMEs.

### 6.4.2. Assessing Cloud BI Operational and Security Functionality

This component allows SMEs to conduct some assessments with each identified Cloud BI application to determine the effectiveness of operational and security functionalities based on the initial risk analysis and enterprise expectations. The framework prioritises the assessment of data management functionalities such as data visualisation, dashboards, and reports that enable SMEs to manipulate data and present results in formats easily understood by non-IT specialists. SMEs should assess the ease of Cloud BI integration with existing enterprise information systems to avoid costly disruptions due to incompatibility which may lead to the unavailability of assets [7,63]. Therefore, the framework suggests that SMEs should assess security functionalities and controls in Cloud BI to verify their effectiveness. It is prudent for SMEs to assess security vulnerabilities in the Cloud BI and potential threats and risks they pose to the existing information system [37,64]. SMEs require information on whether the Cloud BI that they intend to adopt meets their expectations about data and application security, the presence and effectiveness of security controls, and the implementation of security procedures by CSPs and their employees [29]. This implies that SMEs have to assess access control and authentication, which are also central to the security of an application. A security evaluation framework should assist an enterprise in assessing and evaluating vulnerabilities, threats, and risks in IT systems and recommend the best practices that encourage successful deliveries [64]. Furthermore, evaluation of Cloud BI should consider enterprise data security, the protection of data, applications, platform, service and the infrastructure using a set of policies, technologies, and controls [65].

### 6.4.3. Assessing Cloud Deployment Models' Vulnerabilities, Threats, and Risks

The availability of multitenant cloud deployment models makes this an important component for SMEs to consider when evaluating Cloud BI. Each deployment model has its particular security vulnerabilities, threats, and risks that SMEs ought to be acquainted with before selection. Cloud deployment models provided by various CSPs need to be evaluated and selected based on factors such as security, reliability, performance requirements and existing interdependencies, network costs, and privacy essentials [65,66]. By assessing each deployment model, SMEs will be aware of potential data security breaches and types of threats and risks. Of major importance are the risks arising from inside threats, particularly the CSP's employees and other tenants who can cause security breaches. Furthermore, SMEs will be able to ascertain cloud data and application portability as well as cloud interoperability to avoid data lock-in and incompatibilities. The framework enables SMEs to apply the ISO/IEC19941:2017 for interoperability and portability standards and the NIST cybersecurity framework indirectly, as a recommendation by Mirai Security [18]. Interoperability in multitenant environments is a major issue for enterprises because it ensures the security, reliability, and performance of the systems and CSPs [66]. Similarly, poor interoperability between two different systems can be a major security challenge to data availability to the users [67]; therefore, the security framework encourages SMEs to assess this aspect to ascertain that interoperability between enterprise systems and the Cloud BI is acceptable for their needs. Enterprises need to be aware of the security controls deployed by the CSP in each cloud deployment model and their effectiveness in preventing and mitigating data security breaches and risks [68,69]. This framework encourages SMEs to evaluate cloud deployment models to ascertain cloud portability and interoperability which are key to the security, reliability, and performance of Cloud BI and other cloud services.

### 6.4.4. Assessing Security, Trust and Reliability of CSPs

CSPs play an important role in providing cloud services to tenants and they are, therefore, also considered important for evaluation. Effective cloud service security evaluation processes need pragmatic and flexible use of multiple forms of CSP security posture information [69], something that SMEs hardly have access to. Security, trust, and reliability

of CSPs are important aspects for SMEs to establish beforehand. Checking data security and governance ensures how CSPs secure tenants' data and prevent their exposure to new threats and risks. The CSPs are expected to provide proof of adherence to certification and standards for security, trust, and performance reliability [70,71]. One of the most reliable and recommended ways of assessing whether CSPs adhere to standards and best practices of the industry is through their compliance with well-known standards and quality frameworks [33,34]. This framework enables SMEs to assess CSPs for compliance with standards such as the ISO 27,000 series or possessing recognised and valid certifications as recommended [62,70]. Perkins [58] regards standards as industry or government-approved specifications against which quality can be measured or standardised process documents can be developed to support a specific policy or requirements. The use of standards is a unanimously accepted norm as it provides the basis for comparing a selected security system with a given frame of reference adopted at a national or an international level [72]. The Information Security Forum [63] stresses that an enterprise should check regulatory and compliance requirements that can be met by CSPs. Similarly, Tofan [73] argues that, when requested, CSPs are compelled to show to their clients the documented policies and procedures that they use.

Reports on security breaches can be of help to SMEs when assessing CSPs in providing security and privacy to data they keep on behalf of clients [70]. SMEs should be certain about their security responsibilities regarding their data and applications migrated to the cloud. The reliability of service provision by CSPs can be ascertained from reports on the duration of downtime and mechanisms put in place to compensate the clients for business loss. This also applies to the level of data control the enterprise has after migrating to the cloud. On the business side, adherence to contracts and SLAs by the CSPs is very important for SMEs to verify to avoid conflicts in the future [64]. Assessing the business standing of a CSP enables the SMEs to verify whether the provider can continue in the business for a longer period. Besides technical infrastructure security and data security, Vacca [74] encourages cloud clients to understand standards and procedures about the cloud service they want to adopt and how they are implemented by CSPs.

6.4.5. Assessing the Potential Financial Risks of Using Cloud BI

SMEs are generally concerned with making profits by minimising unnecessary financial expenses that might arise from the unbudgeted expenditure. The framework identifies service disruption due to unplanned downtime, hidden costs of subscription, litigation costs, and penalty costs due to misuse of services as additional aspects to assess in the process of security evaluation. The literature confirms that financial risks can arise from additional costs where the implementation might not be smooth; there is a risk of disruption to the organisation and customers; customers are lost to competitors; there is a risk of exposure of the organisation in terms of security and what could happen if complex network management systems malfunction [7,75]. Besides the cost of acquiring the Cloud BI, the proposed framework encourages the assessment of financial risks that may arise from unplanned downtime of service which disrupts business operations. The National Computing Centre Group [76] regards cloud service downtime as a top concern for enterprises and one of the major reasons for customers not willing to adopt and use cloud-hosted software. Enterprises using Cloud BI depend on CSPs for most of their business transactions and can suffer financial risks if the service is down, unavailable, or unreliable [77]. According to Ereth and Dahl [77], the performance of Cloud BI in an enterprise depends on the performance and reliability of the CSP; therefore, client enterprises should guard against paying for poor-performing CSPs. By assessing this aspect, SMEs can make informed decisions about the costs likely to be incurred when they adopt and use Cloud BI.

Another potential source of financial risks is the hidden costs arising from undue subscriptions charged outside the contracts as well as penalty costs for misuse of services. Hidden costs are more prevalent in public than other cloud deployment models [75]. To

avoid hidden costs, enterprises are encouraged to assess if different types of services are available, including on-demand, reserved, or spot instances and the relevant storage, networking, and security required to match the enterprise's workload, requirements, and expectations [29]. The literature shows that some CSPs pretend to offer free assistance to clients and later charge on the expiry of the trial period or when the client decides to move data and applications out of the cloud to another CSP [77]. CSPs achieve this by making the initial subscription free or low to lure unsuspecting potential clients and then increase it sharply after the trial period or after adoption [75].

The need to assess potential litigation costs if clients' sensitive data are exposed to the public is also important for public and community clouds. Litigation costs are unexpected and arise from non-compliance by both the CSP and client enterprise that attracts lawsuits by customers or penalties by governments in the country where the data are hosted [76]. Regardless of the best measures put in place by CSPs, enterprises can end up paying expensive lawsuits for failing to protect sensitive customer information from cybersecurity threats that breach privacy [28]. This implies that SMEs should verify if the CSP once faced litigation at some point in time.

Different CSPs use different pricing models which the SMEs should check to avoid running into unnecessary costly financial risks [36]. In this context, decision-makers must analyse the items paid for with each Cloud BI. For the rented Cloud BI, SMEs need to assess financial risks which may include contract modification or cancellation fees, the additional overhead of managing CSPs, and penalties from overuse of services [78]. The conceptualised framework can help SMEs to achieve this.

The fear of incurring penalty costs for misusing cloud facilities can cause SMEs to avoid the adoption and utilisation of Cloud BI. An enterprise is expected to comply with regulations and standards regardless of where the data are stored, and failure to do so can lead to penalties by CSPs or local authorities [75]. The enterprise has to assess the potential penalties resulting from non-compliance and not turning on some services [79]. This implies that SMEs should take into consideration the possibilities of being charged extra costs for using applications they have not subscribed to.

### 6.4.6. Checklists for Use in the Evaluation of Cloud BI Applications

Seven checklists were provided as a means by which decision-makers will implement the framework. Several studies report checklists as a basic means of providing users with guidelines and criteria to implement a framework [45,54,61]. A study by Müller and Rood [55] encourages the use of checklists as guidelines to assess IT systems. When designing frameworks, researchers are encouraged to provide frameworks to guide the users of the framework on the implementation and metrics needed [54,55]. The checklists derived from the framework were validated as acceptable by SME decision-makers and IT security specialists. When designing the checklists, several factors were considered, including content, layout, language, and metrics to be used [53]. Similarly, the authors in [45,54] posit that an overlap of criteria items in checklists provides consistency in supporting the evaluation of key issues. Respondents confirmed that the checklists contain the basic ideas needed by decision-makers to embark on Cloud BI application evaluation on their own. The advantage of checklists in this study was that the majority of the SMEs were already familiar with their use as evaluation tools. Well-designed checklists should be easy to read and contain simple metrics for use by different users [56]. Due to the wide use of checklists by different enterprises, Müller and Rood [55] argue that users are free to modify them to suit their needs. The checklists provided in this study were meant to enable decision-makers to understand and conduct the process of security evaluation for Cloud BI application at their own pace and within their financial means [56].

### 7. Conclusions

The study found that the majority of the SMEs surveyed do not use industry standards and policies or best practice when evaluating Cloud BI applications due to the lack of

relevant security knowledge and skills needed to use these tools. Existing frameworks and standards present challenges to SMEs with limited knowledge to evaluate Cloud BI because they apply to a large business IT environment with security specialists. The study concluded SMEs require user-friendly and cost-effective security frameworks to properly evaluate the Cloud BI applications they need to adopt. SMEs were prepared to use simple tools and methodologies within their IT knowledge and skills.

There was a significant association between the components and their relevance validation at $p < 0.05$. The acceptance level of the framework and checklists among IT specialists and decision-makers was high. The study concluded that the five main components of the conceptualised security evaluation framework were relevant and acceptable as they addressed basic important issues to be evaluated when selecting Cloud BI applications. The conceptualised framework enabled decision-makers to utilise the best practices from existing traditional industry standards and frameworks.

The limitation of this study is that the findings cannot be generalised to SMEs in different provinces due to differences in economic setup and business expectations, as well as the technological aptitude of users. Further studies will be conducted on how the conceptualised framework impacts the evaluation of Cloud BI applications by SMEs in other provinces in South Africa.

**Author Contributions:** The conceptualisation, methodology, validation, formal analysis, investigation, resources, data curation and writing and original draft preparation were all done by M.M. Review, editing and supervision were done by M.L. The authors of this article are M.M. and M.L., with the former contributing 85% of the work reported. All authors have read and agreed to the published version of the manuscript.

**Funding:** This research received no external funding.

**Institutional Review Board Statement:** The study was conducted according to the guidelines of the Declaration of Helsinki, and approved by the Institutional Review Board (or Ethics Committee) of UNIVERSITY OF SOUTH AFRICA (protocol code ERC Reference #: 014/MM/2016/CSET_SOC 16 February 2016 and Updated 26 June 2020).

**Informed Consent Statement:** Informed consent was obtained from all subjects involved in the study prior to sending the questionnaires and also a statement was included on the questionnaire. Written informed consent has been obtained from the patient(s) to publish this paper. This a requirement stipulated in the Ethical clearance granted by the University.

**Data Availability Statement:** The data presented in this study are available on request from the corresponding author. The data are not publicly available due to [ethical issues with participants. In the informed consent, we had not asked for permission to make the data available to the public].

**Conflicts of Interest:** The authors declare no conflict of interest.

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
