# Peer review of "Conceptualising a Cloud Business Intelligence Security Evaluation Framework for Small and Medium Enterprises in Small Towns of the Limpopo Province, South Africa"

_information, doi:10.3390/info12030128_

Round 1

Reviewer 1 Report

The goal of this study was to investigate the security evaluation practices among SMEs in small South African towns when adopting cloud business Intelligence. The concept of the article respects the general IMRD schema for research articles. Introduction is worked on the acceptable level out and it also presents the description of the state of art in cloud business tools. The methodological part of the contribution presents the technique of collecting data. The methodology is also described correct and sufficient. Also three research questions are formulated for this article.

Results and presentation of data is correct included the proposal of conceptual framework and its validation. All RQs are answered and relatively deep discussed.

Questions for authors: This article is focused on the problems of adopting of cloud technology in SMEs in small towns in the South Africa. You have presented 57 respondents of Your questionnaire. Because I am not very familiar with South African economy, and I think that the majority of readers as well, cloud You please add a short description of this segment of the economy – What does mean the term “small town”? I know that there is a different interpretation of this term for example in China and in Luxemburg. Further - How many SMEs are in South Africa (Limpopo province) and in small towns? Results of Your survey are only from Limpopo province? What is the character of the province´s economy. Based on this information, please add a short section with limit of Your results and conclusions.

How is the sample of 57 respondents representative for Limpopo province´s economy (SMEs and Small towns)?

Author Response

Is the research design appropriate?

Response #1: Justification for use of a small sample due to lack of a sample frame. There number of SMEs using IT systems to support business operations is not documented.

Are the results presented?

Response #2:Tables were reworked. More results were provided for the framework and checklist evaluation. Checklists were provided as Figure 8 to 13

What does mean the term “small town”?

Response #3: In South Africa, we have cities, towns, townships and locations. However, the term township has a political connotation and we decided to use “Small towns”. Small towns are commercial centres for municipalities. The populations of small towns can range from 100 000 to 200000 people.

How many SMEs are in South Africa (Limpopo province) and in small towns?

Response #4: The number of registered SMEs in 9 South African provinces has been included for the 3rd quarter of 2019 as this is the only available record.  Table 1 on page 2. This information had been excluded to focus on cloud security.  Information about SMEs in different small towns in Limpopo is also provided under methodology. Unlike the national, registered SMEs Provincially, detailed information about each town in provinces remains sketch and this makes it difficult to have a sample frame for randomised samples. This is why we relied on network samples.

EDITIONS: Line 44 to 55 (pg 1-2). The number of registered SMEs in the nine South Africa provinces during the 3rd quarter of 2019 was estimated to be close to seven hundred and eighty-seven thousand (787300) [11],[12], shown in Table 1

Table 1: Distribution of SMEs in the 9 South African Provinces

Province

Estimated of SMEs

%

Gauteng

277917

35.3

KwaZulu-Natal

154311

19.6

Western Cape

124393

15.8

Mpumalanga

61409

7.8

Limpopo

48025

6.1

Eastern Cape

41727

5.3

North West

36216

4.6

Free State

33854

4.3

Northern Cape

9448

1.2

Total

787300

100

The information in Table 1 shows that 71% of the SMEs are mainly found in three most developed South African provinces namely; Gauteng (35.3%); KwaZulu-Natal (19.6%) and Western Cape (15.8%) while the other six provinces account for less than 30%. Provinces such as Mpumalanga, Limpopo, Eastern Cape, North West and Free State are mainly and have a single administrative city while most of the settlements are small towns pursuing different commercial activities [11],[12]. Our challenge is finding enterprises using IT system to support business operations hence the use of the snowballing technique in locating them.

What is the character of the province´s economy? Based on this information?

Response #5: This part was included starting with edition, lines 44 to 55 page then on the methodology for Limpopo province. Farming, mining, tourism, trade and commerce are the main economic activities in Limpopo

How is the sample of 57 respondent’s representative for Limpopo province´s economy (SMEs and Small towns)?

Response #6:  The study focuses on SMEs currently using IT system and are making effort to adopt and use Cloud BI applications. This study was limited to those SMEs in the Limpopo Province only. We got information about each enterprise through referrals by other enterprise and also internet search. There is no official database for SMEs using IT and online systems to support business operations. Instead of sending questions to all SMEs, we had to target the few ones who met our inclusion criterion. For the purpose of this study, we think the sample was good enough to give us insights into what is taking place. However, in future, we will expand to other provinces or towns.  In South Africa, there are big towns designated as cities and different towns depending on the economic activities such as mining, farming, rural and tourist, are usually designated as small towns to distinguish them from villages and townships. The population of a small town can range from 6 000 to 16 00 people. Instead of using names of different towns, and their economic activities we preferred to adopt the generic term “small towns”.

Number of small towns in Limpopo

Response #7:  The number of small towns in Limpopo was given as 44 and one cities. Only the number of SMEs in Polokwane is known while those in each small towns are not documented. However, the estimated numbers of SMEs in each category is given for 2019. No data for 2020. 

Reviewer 2 Report

This article proposes a framework to assist SMEs in a South African region in selecting appropriate cloud business intelligence solutions. The proposed framework has been developed based on the results of a questionnaire filled by 57 decision makers of SMEs in the focus region. The article provides details on the conducted questionnaire and findings obtained from the questionnaire. Furthermore, the article discusses relevant elements of the proposed framework by means of three research questions.

The first 4 sections of the article are solid. The topic is introduced appropriately in Section 1. Some relevant background information is provided in Section 2. Section 3 introduces the methods applied to conduct the questionnaire. Finally, results obtained from the questionnaire are described and illustrated appropriately in Section 4. Overall, the first four sections are convincing.

However, I see several issues with the remainder of the article, starting from Section 5:

  • First, I was surprised that the article did not conclude with the results from the conducted questionnaire, since Section 3 (“Methods”) focused on the questionnaire only. From Section 5 I learned that the questionnaire was actually just the first step. This should be reflected by Section 3.
  • I am not really convinced by the method applied to derive the proposed framework. First, the article does not explicitly describe the method applied to derive the framework (as mentioned above, this should be described in Section 3). Second, it seems that the elements of the framework are derived from best practices currently applied by the SMEs, which have filled the questionnaire. This seems a bit strange. In the beginning you state that most SMEs are stuck in the evaluation phase and hence unable to choose appropriate cloud BI solutions. But then you simply take the best practices applied by these SMEs to derive a framework that should help them to accomplish this task. In other words: Given the unsuccessfulness of the SMEs in choosing the right cloud solutions, I doubt that their current best practices are a useful input for a framework that shall help them to choose cloud BI solutions. Maybe their best practices are not good at all (otherwise they would be more successful)? I guess it would be OK to derive *requirements* for the framework from the questionnaires, but I doubt that best practices collected via the questionnaires can directly be used as the framework’s building blocks.
  • The evaluation of the proposed framework (Section 5.2) is rather short does not provide many details. Hence, I am not really convinced that the framework actually does what it is supposed to do.
  • I am somehow missing some information on how the proposed framework can be applied in practice by an SME. What is the SME actually provided with? Is it just the figure and the bullet list from Figure 7? Is this sufficient for an SME to make an educated choice?
  • I am also not so happy with Section 6 and the discussion of the three research questions. All research questions can be answered directly from the results of the questionnaire. So, if it was the goal to answer the three research questions, the entire Section 5 (which is anyhow the weakest part of the article) can be omitted completely. If Section 5 and the proposed framework are considered a relevant part of the article, this framework should somehow be covered by a research question.

Overall, I think the article has a solid core, i.e., the first 4 sections and Section 6 (if you ignore the proposed framework). Of course, the general validity of the results is limited, as focus is put on a certain region in South Africa. Still, the results presented in Section 4 are convincing. However, the subsequent introduction of the proposed framework causes various problems:

  • The method applied to derive the framework is not convincing, as the framework is basically a collection of current best (?) practices of the interviewed SMEs.
  • The validation (evaluation) of the proposed framework is poor.
  • The framework is not covered by any research question.
  • It remains unclear how the framework is applied in practice.

Hence, this part of the article needs to be revised thoroughly before publication of this article can be recommended. Despite these major flaws, there also some minor issues left:

  • The article still contains some typos
  • Table 3 is again followed by Table 2 (i.e., there are two “Table 2”)
  • The text refers to a “Table 5” which is not part of the article
  • Table rows are hard to recognize – maybe some horizontal lines or vertical spacing would help

Author Response

REVIEW 2

_

Is the research design appropriate

Response #1: Has been reworked to justify the use of the sample size and sampling methods.

Are the methods adequately described?

Response #2: Method of data collection and processing have been also  improved by describing further how we collected data and proposed the framework

Are the conclusions supported by the results?

Response #3: Reference has been made to the results in the conclusion. We stated our conclusion based of the findings for question 1 to 3.

If Section 5 and the proposed framework are considered a relevant part of the article, this framework should somehow be covered by a research question.

Response #4: A research question has been added to cover the framework. What can be the main components of a security evaluating framework for Cloud BI applications suitable for SMEs in small towns?

I am somehow missing some information on how the proposed framework can be applied in practice by an SME. What is the SME actually provided with? Is it just the figure and the bullet list from Figure 7? Is this sufficient for an SME to make an educated choice?

Response #5: This framework is accompanied with data checklist for metric purposes that SMEs can use for this purpose. A checklist will be provided for use with each component

First, I was surprised that the article did not conclude with the results from the conducted questionnaire, since Section 3 (“Methods”) focused on the questionnaire only

Response #6:  We have included a research question for the framework which was the core business of the paper and an explanation in the research method

. From Section 5 I learned that the questionnaire was actually just the first step. This should be reflected by Section 3.

Response #7: We have included information on this issue in the Methodology section

The method applied to derive the framework is not convincing, as the framework is basically a collection of current best (?) practices of the interviewed SMEs. The validation (evaluation) of the proposed framework is poor.

Response #8: The best practices are not from the SMEs, but from the various literature sources. We only wanted to establish acceptable to decision makers and whether could be used for a framework. SMEs did not use any of these best practices as they faced challenges. SMEs were requested to implement the framework and questionnaires. Questions for evaluations were based on the practical implementation of different aspects of the framework via questionnaires

The framework is not covered by any research question.

Response #9:  RQ4 has been provided to cover this aspect. What are the main comments of a security evaluation framework for Cloud BI suitable for SMEs in the Limpopo province

We realised that the section for checklist was not included in the first draft. That section has been included. 5 hecklists derived from the framework components. Simple and brief explanations how users can use the framework have been included

The evaluation of the proposed framework (Section 5.2) is rather short does not provide many details. Hence, I am not really convinced that the framework actually does what it is supposed to do

Response #10:  The framework evaluation was elaborated. Due to the limitations imposed by the length of the article, some details have to be left out. However, It has been stated that framework was validated by IT and security specialist for it’s relevance. A questionnaire was third questionnaire was used to collect data from these respondents, two weeks after sending the framework and instructions. The is also applies to acceptance

Given the unsuccessfulness of the SMEs in choosing the right cloud solutions, I doubt that their current best practices are a useful input for a framework that shall help them to choose cloud BI solutions.

Response #11:  We wanted to use the best practice of the SMEs who have already adopted CBIs. However, after that SMEs had many challenges, we provided our customised best practices from which SMEs were indicate if they could be able to perform the activity during the evaluation process. I think the way we had expressed the statement gave the reviewer the wrong impression. The best practice referred to were customised from different evaluation tools and some we formulated. Respondents are asked to rate their willingness to use these practices if they were used as components of framework

I guess it would be OK to derive *requirements* for the framework from the questionnaires, but I doubt that best practices collected via the questionnaires can directly be used as the framework’s building blocks

Response #11:  We used the best practices as initially intended because we felt that these would be very easy for SMEs to implement rather having an academic framework which would need a lot of IT knowledge and skills to interpret. As we have alluded to in the introduction and Literature review, the existing frameworks provide effective solutions to the proble we are addressing, but SMEs decision makers have no knowledge, skills and time to interpret and implement conventional frameworks. Our framework is intended to provide disadvantage SMEs with baseline guidelines on how to select cloud BI. Due to this, we asked the respondents to indicate how suitable each of the best practice could be used as a component of the framework. After the reviewer’s comment we realised that we should provide a checklist for the..

The article still contains some typos The text refers to a “Table 5” which is not part of the article

Response #13:  Tables and figures have been renamed as new information was added to the article.  

Table 3 is again followed by Table 2 (i.e., there are two “Table 2”)

Response #14: This was catered for the table has been renamed as required. Due to the addition of some tables, renaming was also done

Table rows are hard to recognize – maybe some horizontal lines or vertical spacing would help

Response #15: I have used the template formats provided. Some tables have been revised to reduce the length and also including repeating top row

Reviewer 3 Report

The authors propose security evaluation framework for evaluating cloud BI applications. The paper is generally well organised and presented. The research questions are properly identified. The framework was conducted and evaluated based on the survey and statistical method. Figure 7 is informative and interesting.

I suggest the authors to improve the paper by the following points.

  1. Incorporate existing standards or regulations related to cloud security into the framework.
  2. Perform a larger scale of experiments (sample size, randomised groups of SME).

Overall, the paper is interesting and well presented.

Author Response

Moderate English changes required
Response #1: Proofreading was done by English specialist

Incorporate existing standards or regulations related to cloud security into the framework.
Response #2: This has been catered for in the checklists provided. Several Items in the five checklists address standards used. Checklist specifically indicates which standards should be checked

Perform a larger scale of experiments (sample size, randomised groups of SME).

Response #3: This is a good suggestion which we think will implement in the next phase of the study in which we want to expand to other towns in the province. However, using randomised samples is difficult because our population is difficult to access as we have very few known interposes using IT systems and cloud applications. We do not want to run the risk of gathering data from respondents who do not meet the inclusion criteria. However, when we move to much-developed provinces, we can apply this. The other problem faced is time. We have conducted the research and nearly exhausted the SMEs in the selected towns. In the future will take this into account to produce results which can be generalised to large populations

Reviewer 4 Report

Below are my main concerns about this article:

  • The authors don’t specify the importance of their study’s results. I don’t see its added value to the research communities, the studied companies, or other companies.
  • The authors mentioned that their main contribution is addressing the security aspects while limiting their options to the methods that don’t require personnel with IT and security backgrounds.
  • The authors have superficially introduced cloud computing and BI, not reflecting how these technologies are improving daily.
  • The authors don’t specify details such as the positions and the education level of the surveyed employees.

The authors should base their study on more recent papers.

I have run the paper through SafeAssign, and it revealed 6% similarities.

Author Response

x) Moderate English changes required
Response #1 We have sent the paper for Editing by two English specialists who assured us that it they have made all necessary corrections. We also used on editing tools including turn-it in

X Background and include all relevant references?
Response #2 Our study is based on the South African Context, the number of studies in this area is limited. Even on a global scale, Very few literatures is specific to Cloud BI security evaluation by SMEs, We have a plethora of literature on general Cloud computing adoption by large enterprises in developed and developing countries. Much of such literature is not relevant to our study. Being a novel technology is South African SMEs, particularly in disadvantaged towns, we took a cautious approach of taking literature for cloud computing and Big data or IoT, Traditional Business intelligence. The literature on the evaluation of Cloud BI by SMEs is very limited in nature. We also tried to reduce the use the open web sources.

X Are the methods adequately described?
Response #3: We have added population and samples of the Limpopo province and also justified why we used network samples. The section for framework proposal was also included in research methods detailing what we did.

Are the results clearly presented?
Response #3: The main challenge we faced when presenting results, was the template used. We were asked to strickly follow the template. However, we reformatted some of the tables so that row heading repeat throughout. We also used different methods of presentation, graphs and tables. It would be useful for the reviewers to identify specific issues we can improve on for the results. We are prepared to go the extra mile to make the paper as acceptable as possible.

Are the conclusions supported by the results?
Response #3: This section was improved by relating our conclusion to the findings made the study on the four research questions and also the validation and acceptance of the framework. If there are any outstanding concerns, we can address them.

The authors don’t specify the importance of their study’s results. I don’t see its added value to the research communities, the studied companies, or other companies.
Response #4: In the South African context, particularly disadvantaged areas, much research is needed in this area. Even at the global stage, the adoption and use of Cloud BI by SMEs still is a grey area which needs to be addressed, particularly in these times where most of the businesses are going on-line and using smart-technologies. Similarly, our study becomes important in addressing literature and methodical gaps in our context.

The authors mentioned that their main contribution is addressing the security aspects while limiting their options to the methods that don’t require personnel with IT and security backgrounds.
Response #5:In this study, our main focus is SME decision-makers who are the key drivers of the South African economy and who experience technological problems. IT and security specialist were also part of the research for validation and also providing ideas of what should be done. We could rather have collected data from IT and security specialist from big companies, and this could have benefited the researchers and NOT the SMEs
who are supposed to participate in using technology to support business operations. We have seen their deficiencies with regards to Cloud BI because of the lack of advanced IT knowledge. We also wanted to avoid being prescriptive by taking the ideas of IT specialists and imposing them on SMEs, a problem similar to existing industry standards and framework which SMEs are expected to use. Instead, we focused on home-grown solutions to solve old problems being faced by modern enterprises. However, when we move into cities, we can use IT-specialists as we deal with bigger enterprises.

The authors have superficially introduced cloud computing and BI, not reflecting how these technologies are improving daily.

Response #5: This problem has been addressed by looking at the benefits and dangers of using these technologies in businesses particularly in enterprises without IT specialist. The context we are writing influenced us especially the technology divide between SMEs and large enterprises. We have acknowledged how cloud computing has brought about changes in the IT industry and how these will continue shortly. During the COVID -19 pandemic, most of South African SMEs using face-to-face business models are closing down or migrate to online. Our research tries to address these dangers that enterprises should avoid when they adopt cloud technologies without evaluation. We have tried to balance our article with comparisons of South Africa and other nations in terms of the adoption of Cloud BI.

The authors don’t specify details such as the positions and the education level of the surveyed employees.

Response #6: We regret having overlooked this fact. I think in future studies we will have to include, we will collect as much demographic data such as educational background as much as possible. For this study, focused own awareness and duration of use of IT systems.

The authors should base their study on more recent papers.

Response #7: We have a big challenge when it comes to literature on Cloud BI security evaluation for adoption by SMEs. While we have a lot of literature on cloud computing from local and international sources, this different from Cloud BI application evaluation. Very little of cloud BI is found related to enterprise information systems. We tried to balance our study by using any useful literature on this topic, as a result, this study will end up contributing to the much-needed literature in cloud BI evaluation.

I have run the paper through SafeAssign, and it revealed 6% similarities.

Response #8: It is is not our intention to have such a high similarity per cent. We realised changes in the reference fields used with IEEE references. We have corrected the reference list and in-text where such discrepancies occurred. Out Turnitin similarity check was lower than you got. We will keep on improving the article to reduce this similarity as much as possible. We also tried as much as possible to use primary sources in our references.

Round 2

Reviewer 2 Report

The current version of the paper is a step forward compared to the initial version. The authors have addressed all comments made in the first review. Overall, the article is OK now.

My only remaining concern is about a potential lack of scientific rigor. To address the comments from the first review, the authors simply added a research question and modified another one. However, no new data has been gathered or additional analyses have been carried out. This is somehow strange.

One would expect that research questions are defined in the very first step. And only based on the defined research questions, an appropriate method is the defined and the research is carried out based on this research. Results from the conducted research can then be used to answer the research questions. This is at least how research is supposed to “work” from my perspective. As you are obviously able to change ex-post large parts of your research questions (adding one, modifying another one) without needing to adapt your data gathering/analysis, one might assume that something is wrong with your research method/design.

In the newly added text, there are still several sentences containing typos and other mistakes, e.g.:

  • "Using the findings from the study and the best practice from existing standards and frameworks were used was conceptualise a security framework…"
  • "Provinces such as Mpumalanga, Limpopo, Eastern Cape, North West and Free State are mainly and have a single administrative city while most of the s settlements are rural towns pursuing different commercial activities"
  • To propose the security evaluation framework, both primary data from SMEs decision-makers, IT and security specialist as well as secondary information on best practice suggested by existing frameworks and standards.

Author Response

Issue #1: Does the introduction provide sufficient background and include all relevant references?

Response #1:  A section “South African Economic sectors” has been included to provide the context of the study. A number of current references have also been used. However, due limited academic research in security evaluation in Cloud BI in South Africa, we relied on reports from government institutions. 

Issue #2: Is the research design appropriate?    
Response #2: The research design has been unbundled to indicate Population and sampling, Instrument design, validation and reliability and data collection. Each section explains what we did and why we did it. You will see that we used collected data at three stages of the research, from SMEs, from IT security specialists (reviewers) or relevance validation and SMEs for acceptance validation. To economise, we briefly explained how consistence and rigor were attended to by discussing how reduced bias from the questionnaires, content and construct validation by IT security specialists from the local university. We pilot tested our three instruments and improved them using the feedback. For reliability, we tested using SPSS variables questionnaire. The overall reliability have Cronbach’s alpha values above 0.6. We accepted these instruments as reliable based on guidelines from previous studies about Cronbach’s alpha values. This is indicator in the literature on Questionnaire design, validity and reliability. To avoid running the risk of making the article too long, we chose to be brief and focused on key issues.

Issue #3: Are the methods adequately described?    
Response #3: We  also made effort in elaborating how sampled and then administered the online questionnaire to our three samples. For SMEs, it was difficult to have a random sample because we did not have enough knowledge about the sample frame and also where the SMEs using online and cloud-services were. We relied on networking samples which allowed us to network with decision makers who had knowledge of other enterprises.  Both data collection methods and framework validation methods were explained in the relevant sections of the methods and also in section five in which the framework was formulated. 

Issue #4: Are the results clearly presented?    
Response #4: As suggested, before, the tables have been improved to make the results readable. Graphs have been reconstructed after the English editor spotted mistakes. 

Issue #5:: Are the conclusions supported by the results?    
T Response #5:  he conclusions have been reworked to remove unnecessary text. We included the conclusions based on the findings particularly answering research questions. 

Issue #6: Alignment of research questions and findings
Response #6: We would like the reviewer to take note that this is a large study in with 6 research questions including the four we presented. The fourth research question being the key one. In our first presentation, we did not think that this was going to be an issue. So we relied on the three research questions. Secondly, we have collected data for framework validation from the relevant sources as we have explained. Our initial plan was to have a Delphi method, but when the COVID-19 pandemic intensified, we resorted to online questionnaires for validation purposes. We agree with your observation, on this issue. We are bound by the Ethics of the University of South Africa on such issues. It was a simple oversight which we will take seriously in our future studies.

Issue #7: "Using the findings from the study and the best practice from existing standards and frameworks were used was conceptualise a security framework…"    

Response #7: The findings from the study and the best practices from existing standards and frameworks were used to conceptualise a security framework, which was then validated by information security specialists and SME decision-makers.

Issue #8: "Provinces such as Mpumalanga, Limpopo, Eastern Cape, North West and Free State are mainly and have a single administrative city while most of the s settlements are rural towns pursuing different commercial activities"    

Response #8: Mpumalanga, Limpopo, the Eastern Cape, North West and the Free State are mainly rural provinces, each consisting of a single administrative city and the rest rural towns with different commercial activities

General comment: The article was edited by an professional editor from the University of South Africa. We corrected all the mistakes which were identified and this included those on the graph, framework and check lists. The paper is now in a better state than it was in the first place. We hope you will notice these drastic improvements. Due to the number of mistakes identified by the edited, we found it difficult to list all of them.

Reviewer 4 Report

Thank you for clarifying the context of your study and addressing most of my comments and concerns.

Author Response

Issue #1: Does the introduction provide sufficient background and include all relevant references?    
Response #1:  A section “South African Economic sectors” has been included to provide the context of the study. A number of current references have also been used. However, due limited academic research in security evaluation in Cloud BI in South Africa, we relied on reports from government institutions.  

Issue #2: Is the research design appropriate?    
Response #2: The research design has been reworked to indicate Population and sampling, Instrument design, validation and reliability and data collection. Each section explains what we did and why we did it. 

Issue #3: Are the methods adequately described?    
Response #3:  Both data collection methods and framework validation methods were explained in the relevant sections of the methods and also in section five in which the framework was formulated 

Issue #4: Are the results clearly presented?    
Response #4:  As suggested, before, the tables have been improved to make the results readable. Graphs have been reconstructed after the English editor spotted mistakes. 

Issue #5: Are the conclusions supported by the results?    
Response 5:  The conclusions have been reworked to remove unnecessary text. We included the conclusions based on the findings particularly answering research questions. 

Round 3

Reviewer 2 Report

The authors have addressed the main remaining concerns I had with the article. Compared to the initial version, the article is now in a much better shape. The significance of content is still limited though, as the article focuses on a very specific area in South Africa. It remains unclear, to which extent the findings are universally valid and can be applied to other regions or even countries. Nevertheless, the article provides a nice case study with some interesting insights. I so not object this article to be published.